# Spatial diversity processing mechanism based on the distributed underwater acoustic communication system

**Manli Zhou** [1], **Hao Zhang**[1,2]*, **Tingting Lv**[1], **Yong Gao**[1], **Yingying Duan**[1]

**1** Department of Electronic Engineering, Ocean University of China, Qingdao, Shandong, China,
**2** Department of Electrical and Computer Engineering, University of Victoria, Victoria, Canada

* zhanghao@ouc.edu.cn

**Data Availability Statement:** All data files are available from the figshare database.(https://doi.org/10.6084/m9.figshare.23995809.v2).

## Abstract

To address the problem of unreliable single-link underwater acoustic communication caused by large signal delays and strong multipath effects in shallow water environments, this paper proposes a distributed underwater acoustic diversity communication system (DUA-DCS). DUA-DCS employs a maneuverable distributed cross-medium buoy network to form multiple distributed, non-coherent, and parallel communication links. In the uplink, a receiving diversity processing mechanism of joint decision feedback equalizer embedded phase-locked loop and maximum signal-to-interference ratio combining (DFE-PLL-MSIRC) is proposed to achieve waveform-level diversity combining of underwater signals. A phase-locked loop module is embedded in each branch of the decision feedback equalizer to eliminate the residual frequency and phase errors after Doppler compensation. Meanwhile, the combining coefficients are determined based on the maximum signal-to-interference ratio criterion, taking into account the residual inter-symbol interference after equalization, resulting in efficient and accurate computation. Additionally, the combined decision values are fed back to the feedback filters in each branch to ensure more accurate feedback output. Simulation and lake experiment results demonstrate that, compared to the single-link communication system, DFE-PLL-MSIRC can achieve a diversity gain of more than 5.2 dB and obtain about 3 dB more diversity gain than the comparison algorithm. And the BER of DFE-PLL-MSIRC can be reduced by at least one order of magnitude, which is lower by at least 0.6 order of magnitude compared to the comparison algorithm. In the downlink, a transmitting diversity processing mechanism of complex orthogonal space-time block coding (COSTBC) is proposed. By utilizing a newly designed generalized complex orthogonal transmission matrix, complete transmission diversity can be achieved at the coding rate of 3/4. Compared to the single-link communication system, the system can achieve a diversity gain of more than 6 dB.

## Introduction

The breadth and depth of research on smart oceans and transparent oceans for ocean development and utilization have been continuously strengthened in recent years. By utilizing the

**Funding:** Hao Zhang was supported in part by the National Natural Science Foundation of China under Grant 91938204. URL of the funder website is 'https://www.nsfc.gov.cn/'. NO, The funders had no role in study design, data collection and analysis, decision to publish, or preparation of the manuscript.

**Competing interests:** The authors have declared that no competing interests exist.

**Abbreviations:** BER, bit error rate; COSTBC, complex orthogonal space-time block code; DAS, distributed antennas system; DFE, decision feedback equalization; DFE-PLL-MSIRC, decision feedback equalization embedded phase-locked loop and maximum signal-to-interference ratio combining; DUA-DCS, distributed underwater acoustic diversity communication system; FBF, feedback filter; FFF, feedforward filter; GSTBC-SM, generalized space-time block coded spatial modulation; MIMO, multiple-input and multiple-output; MISO, multi-input single-output; MRC, maximal ratio combining; MSE, mean square error; MSIRC, maximum signal-to-interference ratio criterion; NSFC, National Natural Science Foundation of China; OFDM, orthogonal frequency division multiplexing; OSTBC, orthogonal space-time block code; PLL, phase-locked loop; Q-OSTBC, quasi-orthogonal space-time block code; RF, radio frequency; RLS, recursive least square; SIMO, single input multiple output; SM, spatial modulation; SNR, signal-to-noise ratio; STBC, space-time block code; STBC-NOMA, space-time block coded non-orthogonal multiple access; TR, time reversal; UASNs, underwater acoustic sensor networks; VCO, voltage-controlled oscillator.

features of broad coverage, fade resistance, and minimal required switching provided by the distributed antenna systems (DAS) [1, 2], underwater sensor nodes collect and transmit data through point-to-point and multipoint-to-multipoint communication to construct a distributed underwater sensor network for the effective transmission of information [3–6].

In shallow water environments, underwater acoustic signals often suffer from significant time delays and strong multipath effects, frequently resulting in unreliable single-link acoustic communications. With the rise of DAS, the demand for diversity combination technology has become increasingly prominent. Diversity is a technique to compensate for channel fading, usually implemented by two or more receiving antennas. The principle is to combine signals from two or more independent and unrelated paths carrying the same information according to different strategies, which can significantly reduce the probability of deep fading at the receiving end and improve the overall signal-to-noise ratio (SNR) of the received signal, thus enhancing the quality of the wireless communication link [7–12].

A detailed study on the optimal diversity combined linear equalizer and the optimal diversity combined decision feedback equalizer (DFE) under known channels was conducted in [13]. In practice, the channel impulse response is often unknown, and the timing synchronization for the signals of each path needs to be conducted. Then people use blind equalization to realize spatial diversity equalization [14, 15]. When the SNR of a single-path signal is lower than the synchronization threshold, the timing synchronization cannot be achieved, which results in a decline in receiving performance. To address this problem, a waveform-level combining algorithm for multi-antenna signals, namely, the multi-antenna signals combining algorithm based on the maximum ratio of frequency-domain components, was proposed [16]. This algorithm implements a weighted combination of each signal in the frequency domain, which improves the SNR and avoids the problem that a single signal cannot complete timing synchronization due to a low SNR. A two-layer recurrent neural network spatial diversity equalizer structure was investigated to replace the traditional spatial diversity equalizer [17]. The structure can fully use training information to obtain a better training effect in the case of a small amount of weight and to improve the performance of the spatial diversity equalizer. In [18], the Maximum Ratio Combining (MRC) diversity algorithm is combined with the Orthogonal Frequency Division Multiplexing (OFDM) system to improve the received signal at the receiver. The primary idea behind MRC diversity is to improve the received signal at the receiver. Besides, spatial diversity and antenna beamforming methods were employed to significantly improve the signal quality performance of wireless networks [19]. A constrained optimization problem is formulated for alleviating the net noise power integrated with the adaptive beamforming vector and ameliorated channel state information. However, most communication technologies initially developed for terrestrial wired and wireless channels do not apply to the underwater acoustic environment [20]. In underwater acoustic communication, spatial diversity is also receiving significant attention as one of the techniques to improve performance under challenging channel conditions. For the joint reception of multiple signals in frequency-selective fading channels, antenna arrays and equalizers are used for diversity reception to improve the quality of wireless communication systems [21]. The multichannel DFE receiver for underwater acoustic communications was proposed [22, 23]. Received signals are processed by a bank of adaptive linear filters that jointly perform matched filtering and feed-forward equalization. Adaptive phase synchronization is then performed on each branch before the signals are combined and passed to a single DFE feedback and decision loop. In [24], the received signal is split into two streams. One stream is equalized after time reversal (TR) combining. Another stream is flipped in the time domain, processed by TR-combining and equalization, and then flipped in the time domain again. After processing, each stream is conducted diversity combining.

At the same time, space-time coding technology can be considered an extension of transmission diversity technology. This technology's outstanding feature is combining source coding technology and transmission diversity technology to improve the communication system capacity and reliability of the communication system. A space-time block code (STBC) technique based on orthogonal emission diversity was proposed by the American scholar S.M. Alamouti, and the most primitive orthogonal space-time block code (OSTBC) was called the Alamouti code [25]. The study framed space–time codes within an optimal SNR framework, demonstrating their ability to attain the maximum SNR [26]. The paper presented a rate 1/2 code for complex symbols which had a smaller delay than the code already known and also presented another rate 3/4 code which was simpler than the one already known, in the sense it did not involve additions or multiplications. A design for a quasi-orthogonal 1-bit code with partial diversity is proposed [27]. The decoder of the proposed codes works with pairs of transmitted symbols instead of single symbols. The proposed transmission scheme [28] combined the benefits of conventional beamforming with those given by OSTBC. Simulation results for a narrow-band system with multiple transmit antennas and one or more receive antennas demonstrate significant gains over conventional methods in a scenario with nonperfect channel knowledge. [29] proposed to use space-time coding and iterative decoding techniques to obtain high data rates and reliability over shallow-water, medium-range underwater acoustic channels. A comprehensive set of experimental results were obtained by processing data collected from real underwater acoustic communications experiments carried out in the Pacific Ocean. The paper demonstrated that by using space-time coding at the transmitter and sophisticated iterative processing at the receiver, we can obtain data rates and spectral efficiencies. The New Jersey Institute of Technology applied the space-time coding technique to acoustic vector sensors and verified the scheme's effectiveness by computer simulation [30]. Northeastern University in Boston has studied the space-time coding scheme combined with OFDM technology, and the reliability of the project is verified by the underwater acoustic communication experiment in New Zealand [31]. In [32], joint time-reversal space-time block coding and adaptive equalization filtered multitone underwater acoustic communication method was proposed. The effectiveness of the proposed method is verified by simulation analysis and real experimental data collected from an indoor pool communication trial. A generalized space-time block coded spatial modulation scheme for open-loop massive multiple-input and multiple-output (MIMO) downlink communication systems was proposed [33]. The information bits are divided into multiple groups with each group modulated by the spatial modulation (SM), where the SM symbols are invoked for OSTBC and quasi-orthogonal STBC structures. Space-time block coded non-orthogonal multiple access (STBC-NOMA) for underwater acoustic sensor networks (UASNs) was proposed to improve reliability by exploiting transmit diversity and spectral efficiency [34]. Results show that STBC-NOMA can significantly enhance the performance of UASNs without the need for prior channel state information status at the transmitter. The details of the comparison table with the related works are provided in S1 Appendix.

Based on the idea of the distributed antenna system, a significant research project of the National Natural Science Foundation of China (NSFC) (the space-sky-ground-sea integrated space information network) has studied the cooperative processing mechanism of distributed underwater acoustic signals for spatial information networks. This paper constructs the general architecture of a distributed underwater acoustic diversity communication system. The system uses geographically dispersed multiple underwater acoustic buoys to provide services for underwater nodes through a collaborative approach and communicates with shore stations in real-time and reliably.

The main contributions of this paper are as follows.

1. The distributed underwater acoustic diversity communication system (DUA-DCS) that we propose can solve the issue of unreliable point-to-point underwater acoustic communication links in shallow water environments and achieve a waveform-level diversity combination of underwater acoustic signals.

2. To input different complex marine environment parameters into the Bellhop model to obtain the underwater acoustic channel models under different shallow seabed topography and sediment environments and calculate the performance of the proposed system in these different marine environments.

3. We propose a receiving diversity processing mechanism of joint decision feedback equalization embedded phase-locked loop and maximum signal-to-interference ratio combining (DFE-PLL-MSIRC), which exhibits the following advantages:

   - To eliminate the residual frequency and phase errors after Doppler compensation, a PLL module is embedded in each branch of the DFE.

   - The combining coefficient of each branch adopts the maximum signal-to-interference ratio criterion, which considers the residual inter-symbol interference after equalization in addition to additive noise interference, leading to efficient and accurate computation.

   - The combined decision output is fed back to the feedback filters in each branch to ensure more accurate feedback output.

   - Considering system performance, complexity, and practical engineering costs, some engineering recommendations are provided.

4. We propose a transmitting diversity processing mechanism of complex orthogonal space-time block code (COSTBC). By utilizing a newly designed generalized complex orthogonal transmission matrix, complete transmission diversity can be achieved at the coding rate of 3/4.

The remainder of this manuscript is organized as follows. The methods sections explain the underwater acoustic channel model, the system architecture and the spatial diversity processing mechanism. Next, the results of the simulations and lake experiments are presented. Finally, the conclusion is given.

## Underwater acoustic channel model

### Bellhop Gaussian beam tracking model

Under the ray theory, the sound energy radiated by the sound source propagates around the sound line. Some sound lines propagate according to a particular path to reach the receiving point, called eigenrays. And the sound field at the receiving point is the superposition result of all these eigenrays. The ray model can calculate a variety of data, such as the sound field's propagation loss and the sound lines' propagation path, which is an effective method for sound field research [35].

In the context of the ray model, the Bellhop model utilizes the Gaussian beam tracking method (Porter & Bunker, 1987) to calculate the sound field in a horizontally non-uniform environment [36]. The underlying concept of the Gaussian beam ray tracing method involves connecting the Gaussian intensity distribution with the central sound line of each Gaussian beam. These sound lines smoothly traverse through the sound shadow region and caustics. The overall structure of the Bellhop model is illustrated in Fig 1 [37]. One can determine the number of multipath, incident angles, amplitudes, and delays by inputting marine

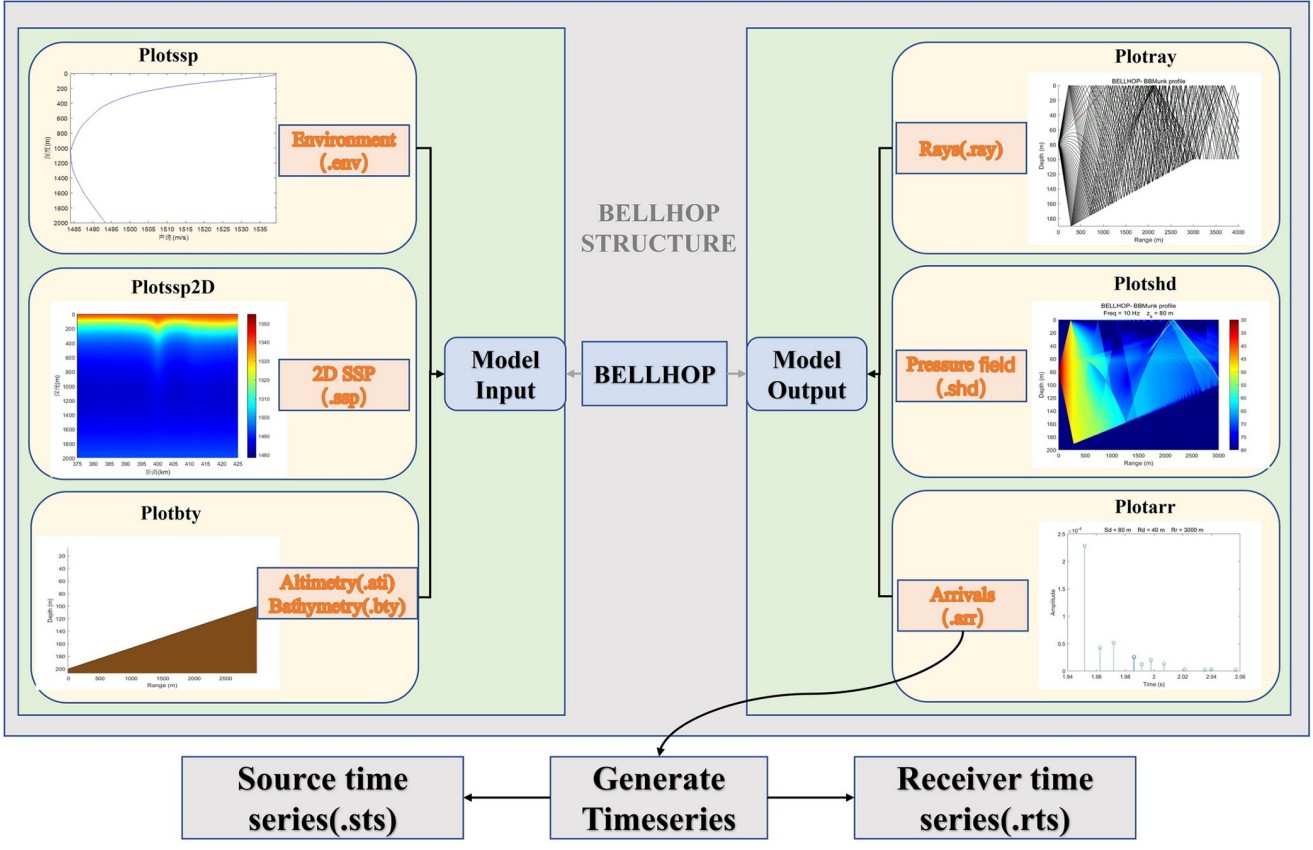

**Fig 1. The structure of the Bellhop.**

environment parameters such as channel geometry, sound speed profile, seabed terrain, and interface reflection loss into the Bellhop model (implemented in MATLAB Bellhop Acoustic Toolbox). These findings serve as valuable references for subsequent adjustments in channel modelling and estimation [38–40].

## Underwater acoustic channel impulse response h(t)

In most scenarios, the underwater acoustic channel can be considered a slowly varying multipath channel with coherence over time. The sound rays emitted from the source travel to the receiver through multiple routes, and the received sound field combines all the arriving rays [41]. Fig 2 depicts a physical model illustrating three propagation paths. It is assumed that each path has the same response amplitude, with $\tau_{21}$ representing the time delay difference between the second and first paths and $\tau_{31}$ representing the time delay difference between the third and first paths. The received signal is the sum of the signals from these three paths.

Assuming that there are $N$ eigenrays, it is considered that the underwater acoustic channel has an $N$ multipath. The impulse response of underwater acoustic multipath channel can be obtainedby the following procedure:

$$h(t) = \sum_{l=1}^{N} A_l \, \delta(t - \tau_l), \tag{1}$$

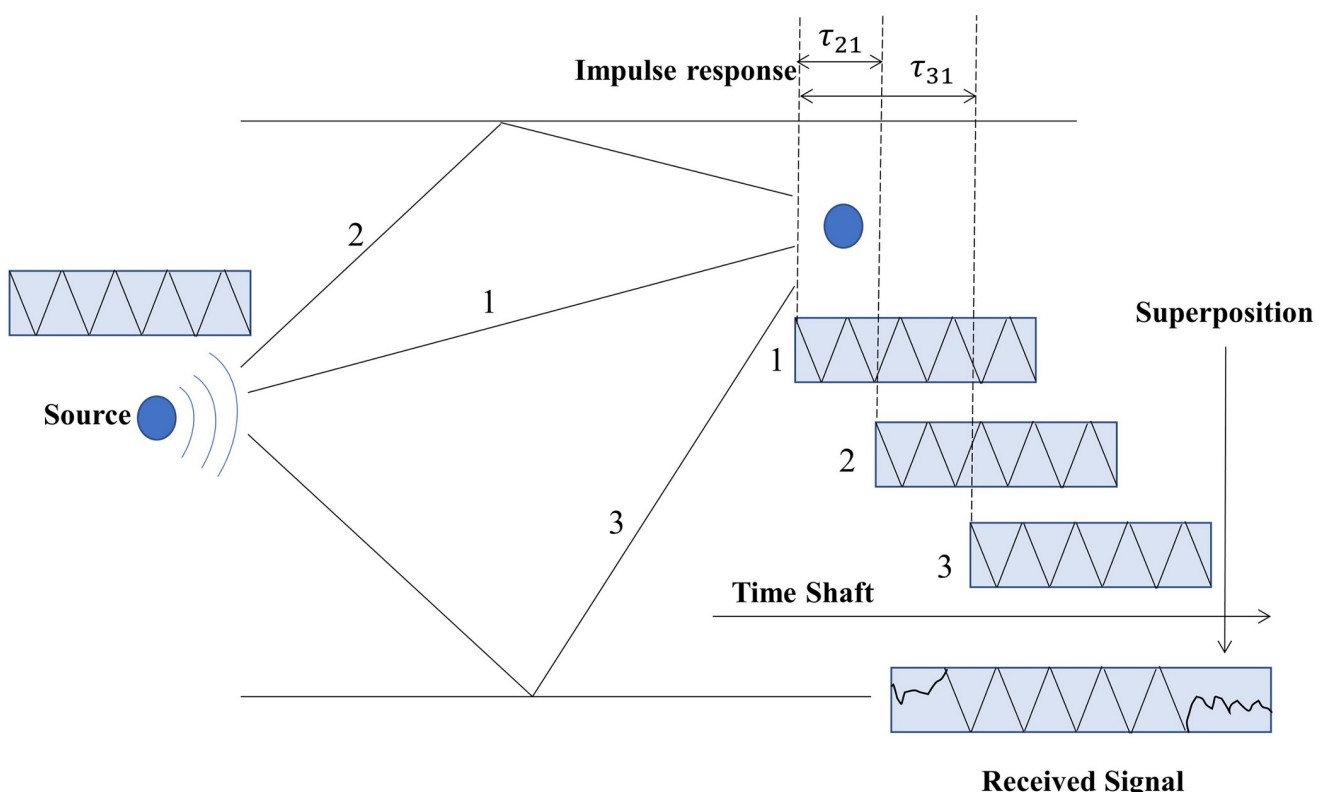

**Fig 2. Physical model of sound line propagation.**

Where, the propagation attenuation coefficient $A_l$ and relative delay $\tau_l$ correspond to different paths in the underwater acoustic channel. Since each path has a different distance, there are additional time delays and energy attenuations for each path to reach the receiving point. The superposition of these rays, with varying time delays and different amplitudes, distorts the received signal waveform.

By providing the underwater acoustic channel parameters, such as the sound speed profile, seabed topography, and sediment density of different marine environments, into the Bellhop model, various outputs can be generated, including transmission loss, eigenrays, and arrival and reception time series. By substituting these parameters into Eq (1), the impulse response $h(t)$ from the sound source to the receiving point can be obtained.

## System architecture

This paper constructs a DUA-DCS based on a cross-media distributed buoys network. Because the multiple underwater acoustic communication links distributed in a certain area have the characteristics of mutual statistical independence, the system uses the buoys network with motorized deployment to convert the 'point-to-point' variable parameter channel into a 'point-to-multipoint' or 'multi-point-to-point' quasi-constant parameter stationary channel. It can form a distributed incoherent parallel communication link. The system adopts the receiving diversity processing mechanism of DFE-PLL-MSIRC and the transmitting diversity processing mechanism of COSTBC to realize the diversity combining of underwater acoustic signals at the waveform level, obtain the signal multipath propagation gain and improve the

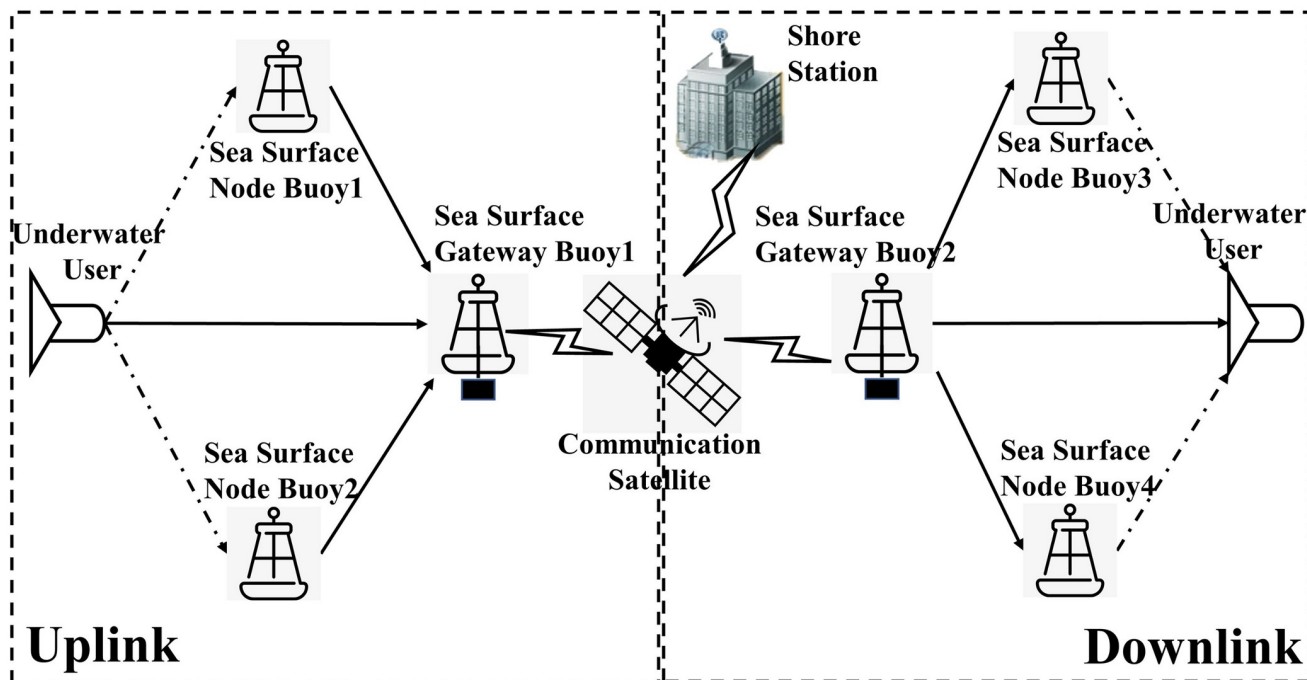

**Fig 3. The architecture of DUA-DCS.**

reliability of data transmission. This system provides real-time and reliable communication between underwater nodes and shore stations. The working mechanism diagram is shown in Fig 3.

The distributed underwater acoustic diversity communication system is divided into uplink cooperative transmission and downlink cooperative transmission.

**The uplink.** The distributed cross-media communication node buoys on the sea surface convert the received underwater acoustic signal to the sampled baseband signal through the underwater acoustic transducer and the radio frequency (RF) front end. Furthermore, the node buoys transmit the signal to the gateway buoy through the ultrashort wave communication network. Then, the gateway buoy processes the underwater acoustic signal received by itself through the underwater acoustic transducer and RF front-end. The processed signal by the gateway buoy and the underwater acoustic sampled baseband signals received from the node buoys are processed in the gateway buoy's underwater acoustic modulation and demodulation module for multi-path waveform-level diversity combining. Finally, the satellite module uploads the combined signal to the shore station. The working mechanism diagram of the uplink is shown in Fig 4.

**The downlink.** The gateway buoy receives the signal sent by the shore station through the satellite module. The system uses the transmitting diversity processing mechanism of COSTBC, and the signal is transmitted in three paths. One path is that the gateway buoy converts the received signal to an underwater acoustic baseband signal through the RF front end and the transducer and sends the baseband signal to the underwater user. The other two paths are that the gateway buoy sends the received signals through the ultrashort wave communication network to the node buoys that can connect to the underwater user. And the node buoys convert the received signal to an underwater acoustic baseband signal through the RF front

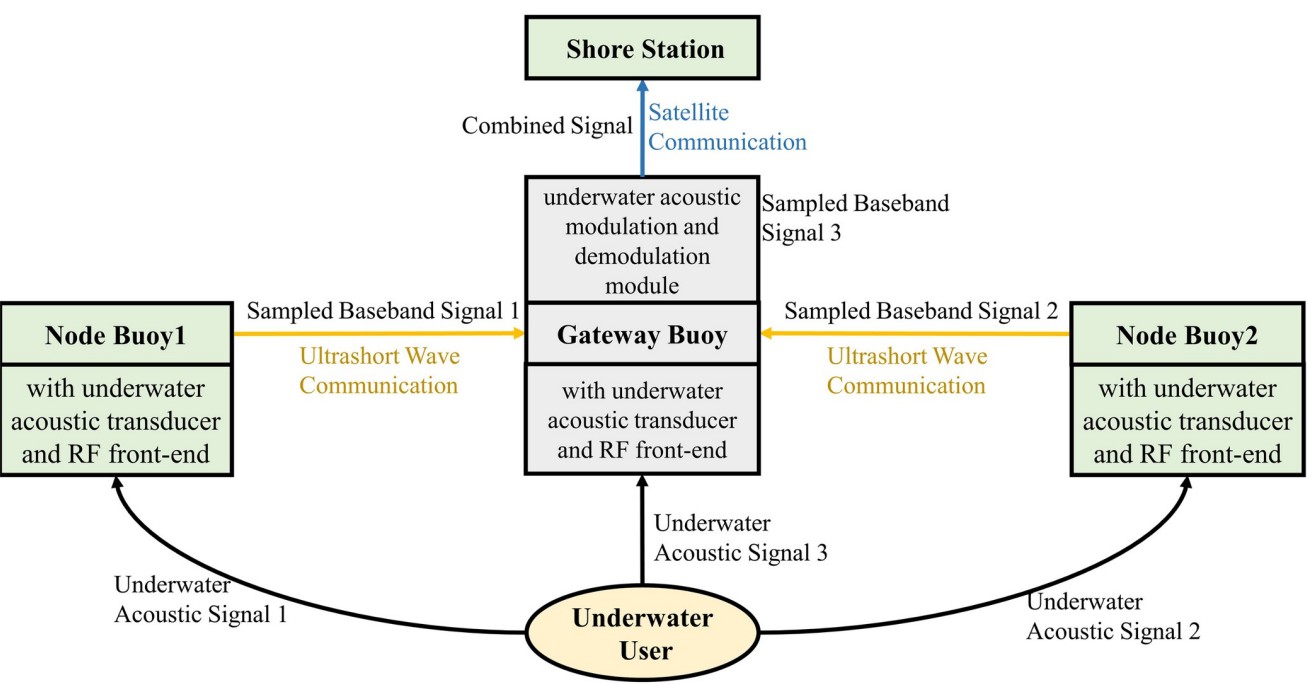

**Fig 4. The working mechanism diagram of the uplink.**

end and the transducer and send the baseband signals to the underwater user. The received signals from multiple paths are processed in the underwater user's underwater acoustic modulation and demodulation module for multi-path waveform-level diversity combining. The working mechanism diagram of the downlink is shown in Fig 5.

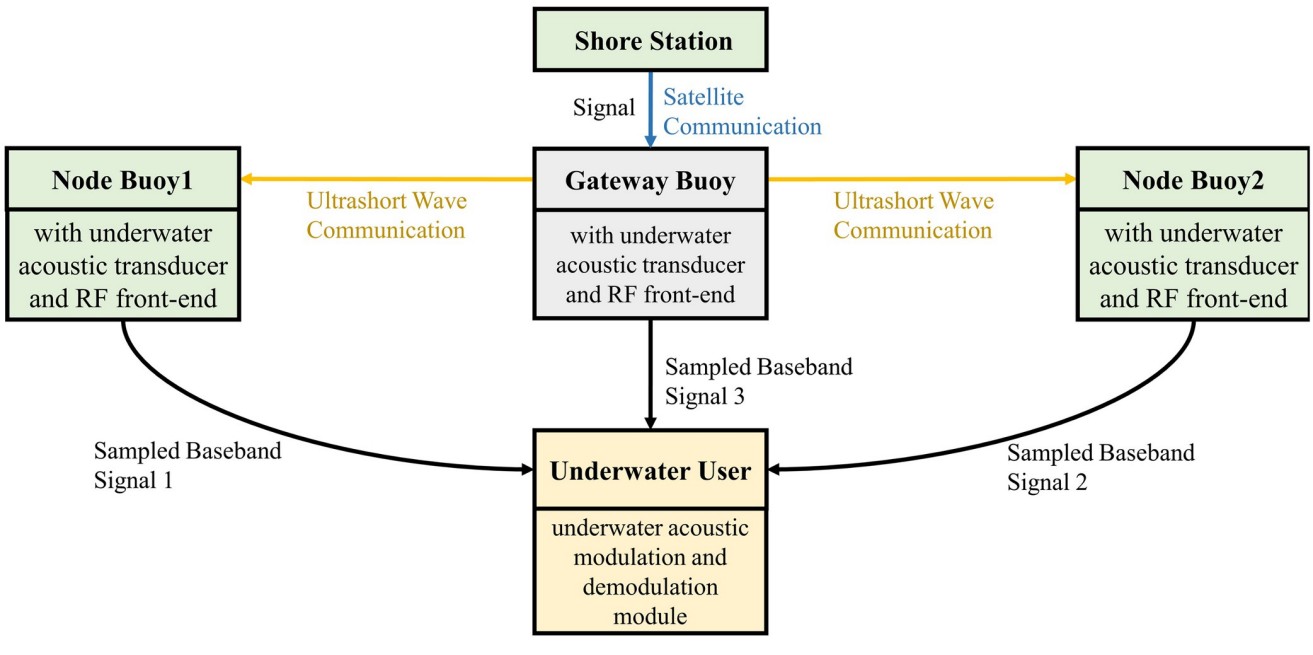

**Fig 5. The working mechanism diagram of the downlink.**

## Spatial diversity processing mechanism

### The receiving diversity processing mechanism of DFE-PLL-MSIRC

**Decision feedback equalizer.** The DFE is a nonlinear time domain equalizer composed of a feedforward filter (FFF) and a feedback filter (FBF). The basic idea of DFE is that once an information symbol is detected and determined, the equalizer can eliminate inter-symbol interference caused by this information symbol before the subsequent symbol is detected. The structure of the DFE is shown in Fig 6.

Assuming that the signal sequence transmitted by the signal transmitter at time instant t is represented as $s(t)$, the digital information sequence of the received signal at the receiver, after passing through the underwater acoustic channel as represented by Eq (1), can be expressed as:

$$x(n) = \sum_{t=0}^{L} h(t)s(n-t) + N(n) \tag{2}$$

where $L$ represents the channel length, $N(n)$ represents the channel's noise. The FBF in Fig 6 is driven by the output of the detector. Its coefficients are continuously adjusted to mitigate the interference of previous symbols on the current symbol. Additionally, it introduces an additional FBF on top of the FFF. This addition effectively suppresses the impact of noise

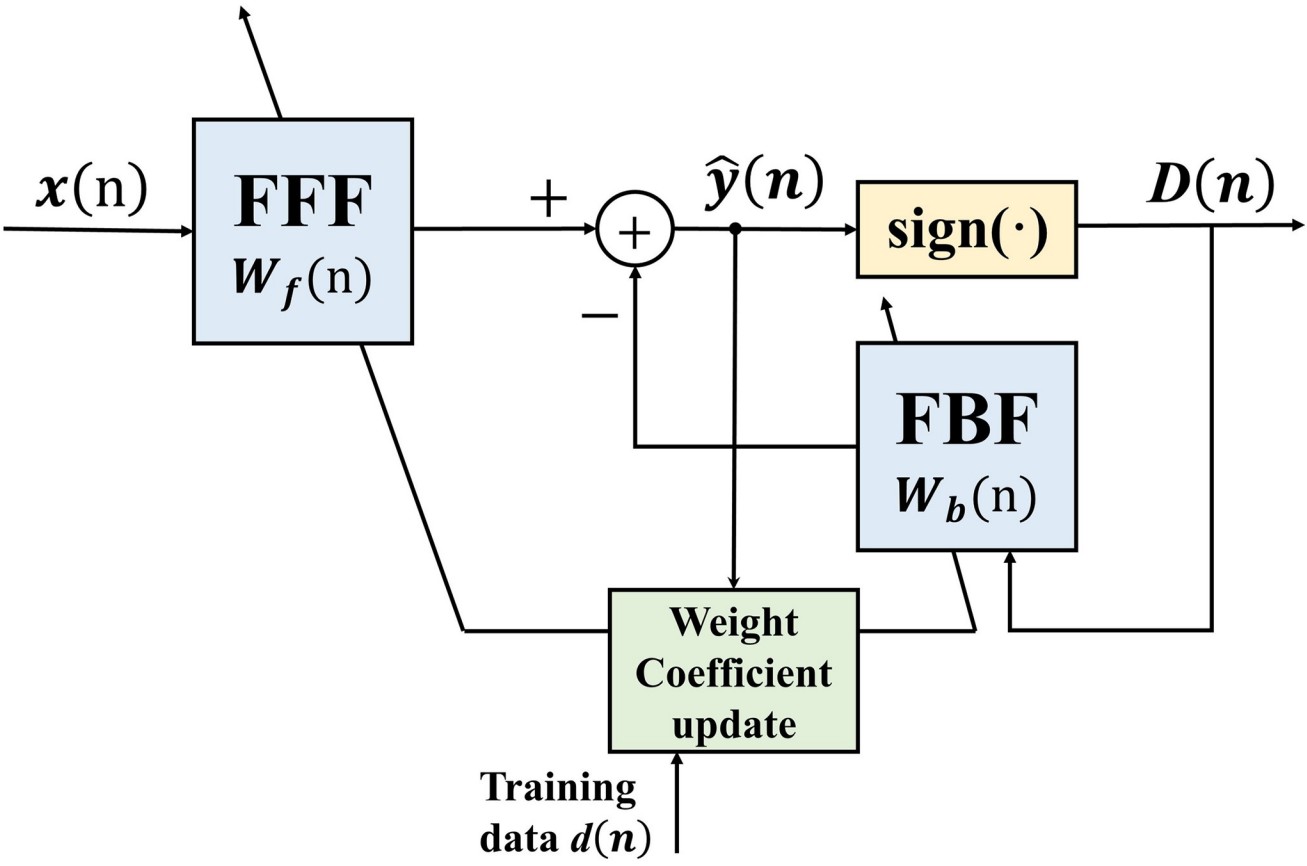

**Fig 6. The structural diagram of the DFE.**

interference on the channel equalization. The output of the FFF is represented as:

$$p(n) = \sum_{k=-N_f}^{0} \mathbf{W}_f^H(k)x(n-k) \tag{3}$$

The output of the FBF is represented as:

$$q(n) = \sum_{j=1}^{N_b} \mathbf{W}_b^H(j)D(n-j) \tag{4}$$

The output of the equalizer is:

$$
\begin{aligned}
\hat{y}(n) &= p(n) - q(n) \\
&= \sum_{k=-N_f}^{0} \mathbf{W}_f^H(k)x(n-k) - \sum_{j=1}^{N_b} \mathbf{W}_b^H(j)D(n-j) \\
&= \mathbf{W}_f^T(n)\mathbf{x}(n) + \mathbf{W}_b^T(n)D(n) \\
&= \begin{bmatrix} \mathbf{W}_f^T(n) & \mathbf{W}_b^T(n) \end{bmatrix} \begin{bmatrix} \mathbf{x}(n) \\ D(n) \end{bmatrix} \\
&= \mathbf{W}^T(n)\mathbf{Z}(n)
\end{aligned}
\tag{5}
$$

The decider can be represented by the following equation:

$$sign(i) = \begin{cases} 1(i > 0) \\ 0(i < 0) \end{cases} \tag{6}$$

The signal after being processed by the decider can be represented as:

$$D(n) = sign(\hat{y}(n)) \tag{7}$$

The output error of the equalizer is represented as:

$$e(n) = d(n) - \hat{y}(n) \tag{8}$$

where $N_f$, $W_f$ and $x(n)$ are the order, tap coefficient and input of the FFF. $N_b$ and $W_b$ are the order and tap coefficient of the FBF. $d(n)$ is the training sequence, which represents the desired output signal. The vector $\mathbf{W}(n)$ represents the FFF's and FBF's combined coefficients. The input vector $\mathbf{x}(n) = [x(n-1), \ldots, x(n-N_f)]^T$ is combined with the decision vector $\mathbf{D}(n) = [d(n-1), \ldots, d(n-N_b)]^T$ to form the input vector $\mathbf{Z}(n) = [\mathbf{x}(n), \mathbf{D}(n)]$ for DFE.

**The recursive least squares adaptive algorithm.** The tap coefficients are iteratively updated by the adaptive recursive least square (RLS) algorithm. The RLS algorithm uses the statistical average of signal errors under the exponential window to correct the tap coefficients of the equalizer. In each iteration, a separate iteration step is given to each tap by inverting the correlation matrix, which speeds up the convergence of the equalizer. The cost equation of the RLS algorithm is:

$$\varepsilon(n) = \sum_{i=1}^{n} \lambda^{n-1}e^2(i) \tag{9}$$

where $\lambda$ is the forgetting factor, which is a tap coefficient that can change the performance of

the equalizer. $e(i)$ denotes the error between the desired output value $d(i)$ of the equalizer and the decision value $\hat{y}(i)$ at the instant i.

The recursive implementation of the RLS algorithm is as follows:

1. Initialization: $\omega(0) = k(0) = 0$, $C(0) = \delta \mathbf{I}_{NN}$, where $\mathbf{I}_{NN}$ is the unit matrix of $\mathbf{N} \times \mathbf{N}$, and $\delta$ is a large number of normal numbers, here we take $\delta = 100$.

2. Recursive calculation by the following equation:

$$e(n) = d(n) - \hat{y}(n) = d(n) - \mathbf{W}^T(n)Z(n) \tag{10}$$

$$k(n) = \frac{C(n-1)Z^*(n)}{\lambda + Z^T C(n-1)Z^*(n)} \tag{11}$$

$$C(n) = \frac{1}{\lambda}[C(n-1) - k(n)Z^T C(n-1)] \tag{12}$$

$$\omega(n) = \omega(n-1) + k(n)e(n) = \omega(n-1) + C(n)Z^*(n)e(n) \tag{13}$$

**Phase-locked loop.** The frequency-selective fading of the underwater acoustic channel, as well as the Doppler shift caused by the relative motion between the transmitter and receiver transducers, can result in phase deviation of the signal, leading to performance degradation of the communication system. Training data can be used to construct a gross Doppler estimate that can be used to resample the data. The Doppler-corrected received signals can then be processed with a DFE with an embedded PLL. Potential instability in the PLL can be overcome by performing phase adjustment prior to the FFF of the DFE. PLL mainly includes a phase detector, loop filter, and voltage-controlled oscillator (VCO). The structure of the PLL is shown in Fig 7.

The input of the phase detector is the voltage $u_R(t)$ corresponding to the input signal and the output voltage $u_V(t)$ of the voltage-controlled oscillator.

$$u_R(t) = U_R \sin[\omega_R t + \theta_R(t)] \tag{14}$$

$$u_V(t) = U_V \cos[\omega_0 t + \theta_V(t)] \tag{15}$$

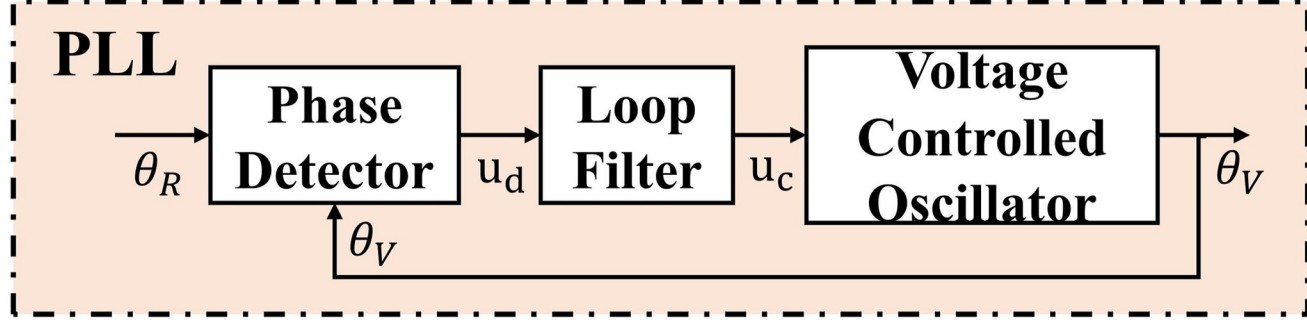

**Fig 7. The structural diagram of the PLL.**

The output of the phase detector can be expressed as:

$$u_d(t) = Ku_R(t)u_V(t) \tag{16}$$

A loop filter is a low-pass filter that filters out high-frequency signals and converts the phase change into a traceable form. The loop filter in the system is implemented using a second-order filter form, and the output of the filter is:

$$u_c(t) = K_1 u_d(t) + K_2 \sum_{i=0}^{t} u_d(i) \tag{17}$$

VCO is essentially an integrator. It is an accumulator for discrete signals. It generates phase output according to the loop filter's input to track the phase change. The output of the voltage-controlled oscillator is:

$$\theta_V(t) = K_V \int_0^t u_c(t)dt \tag{18}$$

where $K_V = K_1 K_2$, $K_1$ and $K_2$ are the coefficients of the proportional and integral parts, respectively. Their selection determines the dynamic range and loop gain of the PLL.

**DFE-PLL-MSIRC.** The application scenario of the uplink of underwater acoustic diversity communication system based on a distributed buoy network proposed in this paper can be seen as the single input multiple output (SIMO) model with one transmitter and three receivers. In the traditional SIMO model, multiple channel output vectors are accumulated together to generate a unified output vector, and the channel equalizer is applied to this long output vector. However, underwater acoustic channels usually encounter very long channel responses. The traditional SIMO model equalization scheme may require high-order channel equalizers (e.g., hundreds of taps) and lead to very slow convergence of tap coefficients. And due to the extensive delay and sparsity of the underwater acoustic channel, the direct superposition of multipath signals may aggravate the signal attenuation, thus causing negative impacts on the system.

Compared with the SIMO-DFE system described in [22, 23], this paper combines spatial diversity techniques with an adaptive decision feedback equalizer embedded PLL at the receiver. Phase-compensated and equalization are performed separately on each branch before diversity combining. Then the output of each equalizer is combined with the maximum signal-to-interference ratio criterion (MSIRC). The combined signal is decided and input into the FBF of each equalizer and the phase detector of each branch, rather than the combined signal itself. The system structure is shown in Fig 8.

The input signal $x_m(n)$ of the m-th branch can be represented as:

$$x_m(n) = \sum_{t=0}^{L} h_m(t)s(n-t) + N_m(n) \tag{19}$$

where $m = 1, 2, \ldots, M$. $M$ represents the number of antennas in the receiver, i.e., the number of diversity branches, and $h_m(t)$ represents the channel impulse response from the transmission transducer to the m-th receiving antenna.

At the receiver end, the expression for the output of the FFF for the m-th receiving antenna is as follows:

$$p_m(n) = \sum_{k=-N_f}^{0} W_{f,m}^H(k)x_m(n-k) \tag{20}$$

When the system uses a PLL to correct residual frequency and phase errors in the received

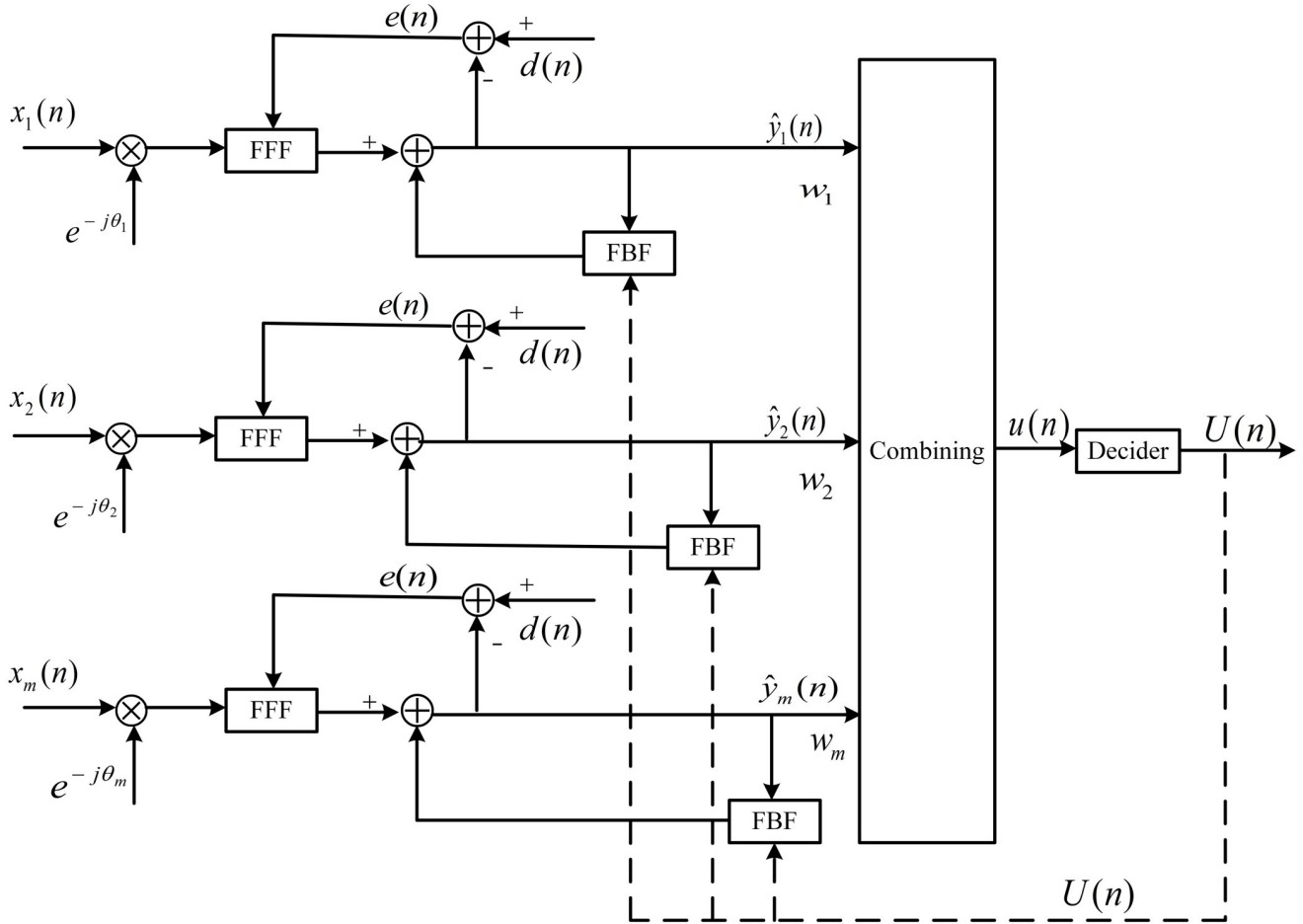

**Fig 8. The structural diagram of SIMO-DFE-PLL-MSIRC.**

signal, the output of the FFF can be expressed as:

$$p_m(n) = \sum_{k=-N_f}^{0} W_{f,m}^H(k)x_m(n-k)e^{-j\theta_m(k)} \tag{21}$$

The output of the FBF for the m-th receiving antenna is represented as:

$$q_m(n) = \sum_{j=1}^{N_b} W_{b,m}^H(j)U(n-j) \tag{22}$$

where $U(n)$ represents the decision value of the combined signal $u(n)$.

The output estimate of the DFE for each m-th branch of the SIMO system can be represented as:

$$\begin{aligned}\hat{y_m}(n) &= p_m(n) - q_m(n) \\ &= \sum_{k=-N_f}^{0} W_{f,m}^H(k)x_m(n-k)e^{-j\theta_m(k)} - \sum_{j=1}^{N_b} W_{b,m}^H(j)U(n-j)\end{aligned} \tag{23}$$

Taking the derivative of the phase $\theta_m(n)$ with respect to $MSE = E|e(n)^2|$, we have:

$$
\begin{aligned}
\frac{\partial MSE}{\partial \theta_m(n)} &= \frac{\partial E\{e(n)|^2\}}{\partial \theta_m(n)} = \frac{E\{|d(n) - \hat{y}_m(n)|^2\}}{\partial \theta_m(n)} \\
&= \frac{\partial E\{d(n) - p_m(n) + q_m(n))(d(n) - p_m(n) + q_m(n))^*\}}{\partial \theta_m} \\
&= -\frac{\partial E\{2 \operatorname{Re}\{p_m(n)[d(n) + q_m(n)]^*\}\}}{\partial \theta_m} + \frac{\partial E\{p_m(n)p_m^*(n)\}}{\partial \theta_m} \\
&= -2 \operatorname{Im}\{E\{p_m(n)[d(n) + q_m(n)]^*\}\} \\
&= -2 \operatorname{Im}\{E\{p_m(n)[p_m(n) + e(n)]^*\}\}
\end{aligned}
\tag{24}
$$

Define $\Phi_m(n)$ as the output of the phase detector:

$$
\Phi_m(n) = \operatorname{Im}\{p_m(n)[p_m(n) + e(n)]^*\}
\tag{25}
$$

The update process for $\theta_m(n)$ based on the instantaneous gradient of $\theta_m(n)$ using a second-order digital PLL and mean square error (MSE) can be described as follows:

$$
\theta_m(n+1) = \theta_m(n) + K_1\Phi_m(n) + K_2\sum_{k=0}^{n}\Phi_m(k)
\tag{26}
$$

As shown in Fig 8, after phase compensation and equalization of each branch of the system, the combining coefficient of each branch adopts the MSIRC. Compared with the MRC, MSIRC takes the residual inter-symbol interference after equalization into account in addition to additive noise interference, which is more reasonable and easier to calculate. The combined decision output y(n), instead of the decision value of a single branch, is input into the FBF and the phase detector of each branch to ensure that a more accurate decision value is fed back to each branch.

The residual inter-symbol interference after equalization in each branch can be represented as:

$$
ISI_m(n) = \hat{y}_m(n) - sign(\hat{y}_m(n))
\tag{27}
$$

The weighting coefficients of each branch are expressed as follows:

$$
w_m(n) = 1/|ISI_m(n)|^2.
\tag{28}
$$

The final signal output by the decider can be expressed as:

$$
u(n) = \sum_{m=1}^{M}\hat{y}_m(n)w_m(n).
\tag{29}
$$

**The transmitting diversity processing mechanism of COSTBC.** The application scenario of the downlink of the underwater acoustic diversity communication system can be seen as a multi-input single-output (MISO) model with three transmitters and one receiver. The input signal is converted into binary code at the three buoy transmitters and performed QPSK modulation. Then, the QPSK signals are transmitted out after COSTBC. The signals are transmitted through three independent underwater acoustic channels. After receiving three signals, the underwater user performs Doppler compensation, equalization on the three signals, and then combination. Finally, the output signal is decoded by the maximum likelihood criterion. In this paper, three antennas are used at the system's transmitter. The signal processing flow of the downlink is shown in Fig 9.

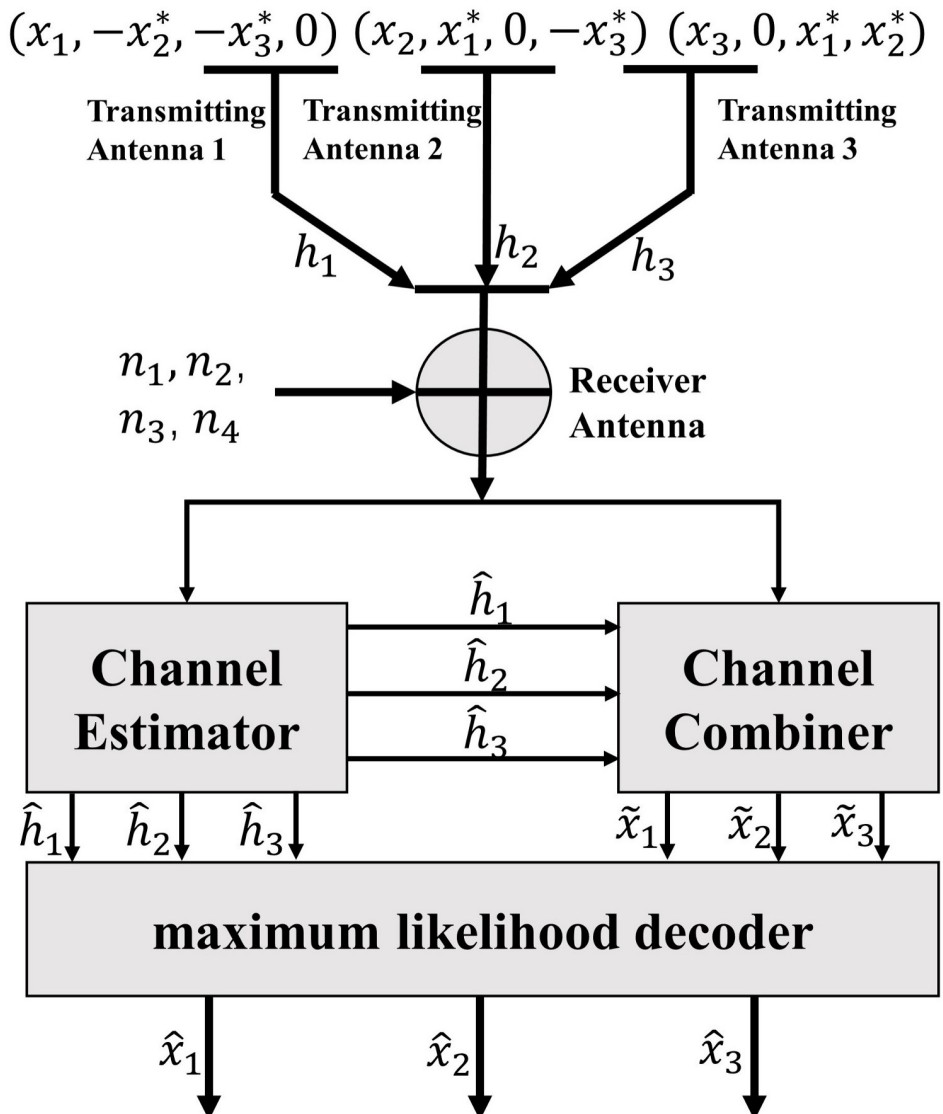

$(x_1, -x_2^*, -x_3^*, 0)\ (x_2, x_1^*, 0, -x_3^*)\ (x_3, 0, x_1^*, x_2^*)$

**Fig 9. The signal processing flow of COSTBC.**

COSTBC is used at the signal transmitter. It is assumed that a constellation with $2^b$ constellation points is used for symbol mapping. Each k information bit is mapped to k modulation symbols$(x_1, x_2, \ldots, x_k)$. STBC takes a set of k modulation symbols in each time slot, which is mapped to the transmission matrix M of $N_t * p$, where p is the number of time slots (i.e., the number of periods required to send a set of coding symbols). Each column is the linear group of modulation symbols transformed in the time domain. The t column of the coding matrix is sent to the $N_t$ transmit antenna at t time. Space-time block coding is encoded both in time and space.

In order to achieve full transmit diversity, transmission matrix $\mathbf{M}_{N_t}$ with $\mathbf{N}_t$ transmitting antennas is constructed based on orthogonal design, and each row of $\mathbf{M}_{N_t}$ is orthogonal. It satisfies Eq (30):

$$\mathbf{M}_{N_t} \cdot \mathbf{M}_{N_t}^T = c\rho \mathbf{I}_{N_t} \tag{30}$$

where $\rho = \sum_{i=1}^{k}|x_i|^2$, $\mathbf{I}_{N_t}$ represents the unit matrix of $\mathbf{N}_t \times \mathbf{N}_t$, and c is a constant. This encoding can provide full transmit diversity at the encoding rate $R = k/p$.

If all nonzero elements of a row in the coding matrix are conjugate, the row is called a conjugate row. Otherwise, it is called a non-conjugate row. Define that the number of rows for all conjugate rows is $v_1$, and the number of rows for non-conjugate rows is $v_0$. Then the design process of COSTBC can be summarized as Algorithm 1:

**Algorithm 1** The design process of coding matrix

**Require**: $\mathbf{M}_1 = x_1 \mathbf{I}_{N_t}$
**Ensure**: $\mathbf{M}_m$, $m$ = 2, 3, ..., $m$
1: Calculation of $v_0$ and $v_1$ in $M_{m-1}$
2: From the first line to the $p_{m-1}$-row, write new elements to the m-column.
3: **if** $v_0 \geq v_1$ **then**
4:   the new element $x_k(k = k_{m-1} + 1, ..., k_{m-1} + v_0)$ is written to the m-column of the non-conjugate row, and zeros are added to the m-column of the corresponding conjugate row.
5:   write $x_k^*(k = 1, 2, ..., k_{m-1})$ into the m-column of $p_{m-1} + 1$ row to $p_{m-1} + k_{m-1}$ row.
6: **else if** $v_0 < v_1$ **then**
7:   the element $x_k^*(k = k_{m-1} + 1, ..., k_{m-1} + v_1)$ is written to the m-column of the conjugate row, and zeros are added to the m-column of the corresponding non-conjugate row.
8:   write $x_k(k = 1, 2, ..., k_{m-1})$ into the m-column of $p_{m-1} + 1$ row to $p_{m-1} + k_{m-1}$ row.
9: **end if**
10: Ensure the orthogonality of each element in the encoding matrix, add new elements to the front m-1 columns of $p_{m-1} + 1$ row to $p_{m-1} + k_{m-1}$ row.
11: Define $row = 1, 2, ..., p_{m-1}$, $col = 1, 2, ..., m-1$, $i = 1, 2, ..., m-1$:
12: **if** $|\mathbf{M}_{m-1}(row, col)| = |x_i|$ and $sign(\mathbf{M}_{m-1}(row, col)) = 1$ **then**
13:   $\mathbf{M}_m(p_{m-1} + i, col) = -\mathbf{M}_m^*(row, m)$
14: **else if** $sign(\mathbf{M}_{m-1}(row, col)) = -1$ **then**
15:   $\mathbf{M}_m(p_{m-1} + i, col) = \mathbf{M}_m^*(row, m)$
16: **else if** $|\mathbf{M}_{m-1}(row, col)| \neq |x_i|$ **then**
17:   $\mathbf{M}_m(p_{m-1} + i, col) = 0$
18: **end if**
19: Ensure the orthogonality of the first m-1 row in the coding matrix, add several row elements and zero at the corresponding position of the m-column.
20: Let $RowCount = p_{m-1} + k_{m-1}$, $row = p_{m-1} + 1, ..., p_{m-1} + k_{m-1}$, $NewRow = p_{m-1} + k_{m-1} + 1, ..., p_{m-1} + k_{m-1} + RowCount$, $1 \leq col1 < col2 \leq m-1$.
21: **if** $\mathbf{M}_m(row, col1) \neq 0$ and $\mathbf{M}_m(row, col2) \neq 0$ **then**
22:   add row elements and set $flag = true$
23:   Let $\mathbf{X}_{11} = \mathbf{M}_m(row, col1)$, $\mathbf{X}_{12} = \mathbf{M}_m(row, col2)$, $\mathbf{X}_{21} = \mathbf{M}_m(NewRow, col1)$, $\mathbf{X}_{22} = \mathbf{M}_m(NewRow, col2)$
24:   $\mathbf{X}_{21} = 0$, $|\mathbf{X}_{22}| = |X_{11}|$, and $\mathbf{X}_{11} = \mathbf{X}_{22}^*$ **then**
25:     $\mathbf{M}_m(NewRow, col1) = -\mathbf{X}_{12}^*$
26:   **else if** $\mathbf{X}_{21} = 0$, $|\mathbf{X}_{22}| = |\mathbf{X}_{11}|$, and $\mathbf{X}_{11} = -\mathbf{X}_{22}^*$ **then**
27:     $\mathbf{M}_m(NewRow, col1) = \mathbf{X}_{12}^*$
28:   **else if** $\mathbf{X}_{22} = 0$, $|\mathbf{X}_{21}| = |\mathbf{X}_{12}|$, and $\mathbf{X}_{21} = \mathbf{X}_{12}^*$ **then**
29:     $\mathbf{M}_m(NewRow, col2) = -X_{11}^*$
30:   **else if** $X_{22} = 0$, $|\mathbf{X}_{21}| = |\mathbf{X}_{12}|$, and $\mathbf{X}_{21} = -\mathbf{X}_{12}^*$ **then**
31:     $\mathbf{M}_m(NewRow, col2) = \mathbf{X}_{11}^*$
32:   **end if**
33: **else if** $|\mathbf{X}_{21}| = |\mathbf{X}_{12}|$ and $|\mathbf{X}_{22}| = |\mathbf{X}_{11}|$ **then**

```
34:    set flag = flase
35: end if
36: When flag = true, set RowCount = RowCount + 1,
    Mₘ(NewRow, col1) = −Mₘ*(row, col2), Mₘ(NewRow, col2) = Mₘ*(row, col1)
37: Let kₘ = kₘ₋₁ + max(v₀, v₁), pₘ = RowCount, repeat the above process
    until m = N.
```

When the number of transmitting antennas is 3, the transmission matrix $\mathbf{M}_1$, $\mathbf{M}_2$ and $\mathbf{M}_3$ in the design process of the above COSTBC are:

$$\mathbf{M}_1 = [x_1] \tag{31}$$

$$\mathbf{M}_2 = \begin{bmatrix} x_1 & x_2 \\ -x_2^* & x_1^* \end{bmatrix} \tag{32}$$

$$\mathbf{M}_3 = \begin{bmatrix} x_1 & x_2 & x_3 \\ -x_2^* & x_1^* & 0 \\ -x_3^* & 0 & x_1^* \\ 0 & -x_3^* & x_2^* \end{bmatrix} \tag{33}$$

The receiver uses maximum likelihood decoding. Assuming the receiver fully recovers the channel fading coefficient and the channel fading coefficient is constant in the p time slot. Namely:

$$h_{j,i}(t) = h_{j,i}(t + T) = h_{j,i}(t) \tag{34}$$

where $t = 1, 2, \ldots, p$.

Then:

$$(\hat{x}_1, \hat{x}_2, \ldots, \hat{x}_k) = arg \min_{\hat{x}_1, \hat{x}_2, \ldots, \hat{x}_k \in S} \sum_{i=1}^{k} [d^2(\tilde{x}_i, \hat{x}_i) + (\sum_{i=1}^{N_i} \sum_{i=1}^{N_t} |h_{j,i}|^2 - 1)(x_i)^2] \tag{35}$$

When the channel parameters are certain, the decision signal $\tilde{x}_i$ is only related to $x_i$. The Eq (35) can be rewritten as:

$$\hat{x}_i = \underset{\hat{x}_i \in S}{argmin} d^2(\tilde{x}_i, \hat{x}_i) + (\sum_{i=1}^{N_i} \sum_{i=1}^{N_t} |h_{j,i}|^2 - 1)(x_i)^2 \tag{36}$$

where $i = 1, 2, \ldots, k$.

## Simulation experiments and results

### Signal transmitter

The system uses QPSK modulation for data transmission. The signal frame format shown in Fig 10 is designed for transmission in the complex underwater acoustic environment. Each data frame contains a Doppler estimation and frame synchronization head, training sequences, data information, and frame interval. Frame intervals are filled with the first 217 bits of data information. This frame format is a short frame format, which has the solid anti-Doppler capability and the ability to eliminate the cumulative timing error.

| | Doppler Estimation and Frame Synchronization Head | Training Sequences | Data Information | Frame Interval |
|---|---|---|---|---|
| Data I | 1023 chips balanced Gold sequences | 1000 chips | 3060 chips | 217 chips of data information |
| Data Q | 1023  chips balanced Gold sequences | 1000 chips | 3060 chips | |

| Data Frame 1 | Data Frame 2 | ... |
|---|---|---|

**Fig 10. The frame format of the QPSK signal.**

## Channel environment

According to sea surface condition, seabed topography, seabed medium, and SNR of the marine environment, we have selected three scenarios of the marine environment, as shown in Table 1. The simulation parameters of the sound source and receiver hydrophones are set as shown in Table 2.

The propagation loss diagram and channel impulse response of each receiving point in three marine environments are shown in Fig 11.

## Simulation results

The complete signal processing flow for underwater acoustic communication is shown in Fig 12. The transmitter converts the input signal into binary code. Then the data are divided into frames, and the frame header and training data are added to each frame. QPSK modulation and up-conversion are performed on the frame-divided data and then sends it out. Underwater acoustic signals transmitted through three independent underwater acoustic channels are received by three node buoys and then aggregated to the gateway buoy for processing. The gateway buoy performs band-pass filtering, down-conversion, bit synchronization, frame header capture, and doppler compensation on the three signals. Then the gateway buoy conducts DFE-PLL-MSIRC and outputs the signal after the decision.

**Case 1**:Verify the effectiveness of the proposed system in this paper and compare it with the proposed system in [22, 23]. The underwater acoustic channel environment is set to 'Marine Environment 1'.

The signals from the underwater acoustic multipath channels are transmitted to the gateway buoy through the node buoys for diversity combining. The simulation results are shown in Fig 13. The number of receiving antennas also represents the number of diversity.

In order to provide a clearer description of the data in the figures, we have compiled the data into Tables 3 and 4.

**Table 1. Classification of marine environment.**

| Marine Environment | Sea Condition | Seabed Topography | Seabed Medium | SNR of Environment |
|---|---|---|---|---|
| 1 | level 2 | Flat | Silty Clay | SNR = 5dB |
| 2 | level 4 | Steep Hill | Sandy Silt | SNR = 0dB |
| 3 | level 7 | 50m Peak | Coarse Sand | SNR=-5dB |

Table 2. Bellhop simulation paramenters setup.

| Parameter | Value |
|---|---|
| Depth of Sea Water | 200 m |
| Depth of Sound Source | 80 m |
| Frequency of Sound Source | 10 kHz |
| Angle of Sound Wave Emitted by Sound Source | $-20° \sim +20°$ |
| Depth of Receiver | 20 m |
| Distance from Receiving Point to Sound Source | 1.9 km, 2 km, 2.2 km, 2.5 km, 2.8 km, 3 km |
| Distance between buoy nodes | 2 km |

From Fig 13, Tables 3 and 4, it can be observed that the system's BER decreases as the number of receiving antennas increases. Compared to the point-to-point communication link system, DFE-PLL-MSIRC can achieve a diversity gain of more than 5.2 dB, while the comparison algorithm can achieve a diversity gain of more than 2.3 dB. The proposed DFE-PLL-MSIRC can obtain about 3 dB more diversity gain than the comparison algorithm. When SNR is 0 dB, the BER of DFE-PLL-MSIRC can be reduced by at least one order of magnitude, while the comparison algorithm can be reduced by at least 0.4 order of magnitude. Overall, the BER of DFE-PLL-MSIRC is lower than that of the comparison algorithm.

When the number of diversity (nR) increases from 1 to 2 and then to 3, the effect of diversity improvement is pronounced. However, with the increasing of the number of diversity, the gain obtained by diversity does not increase too much. Considering the performance, the complexity of the system, and the cost of practical engineering, we recommend setting the number of diversity of DUA-DCS to 3 in practical engineering.

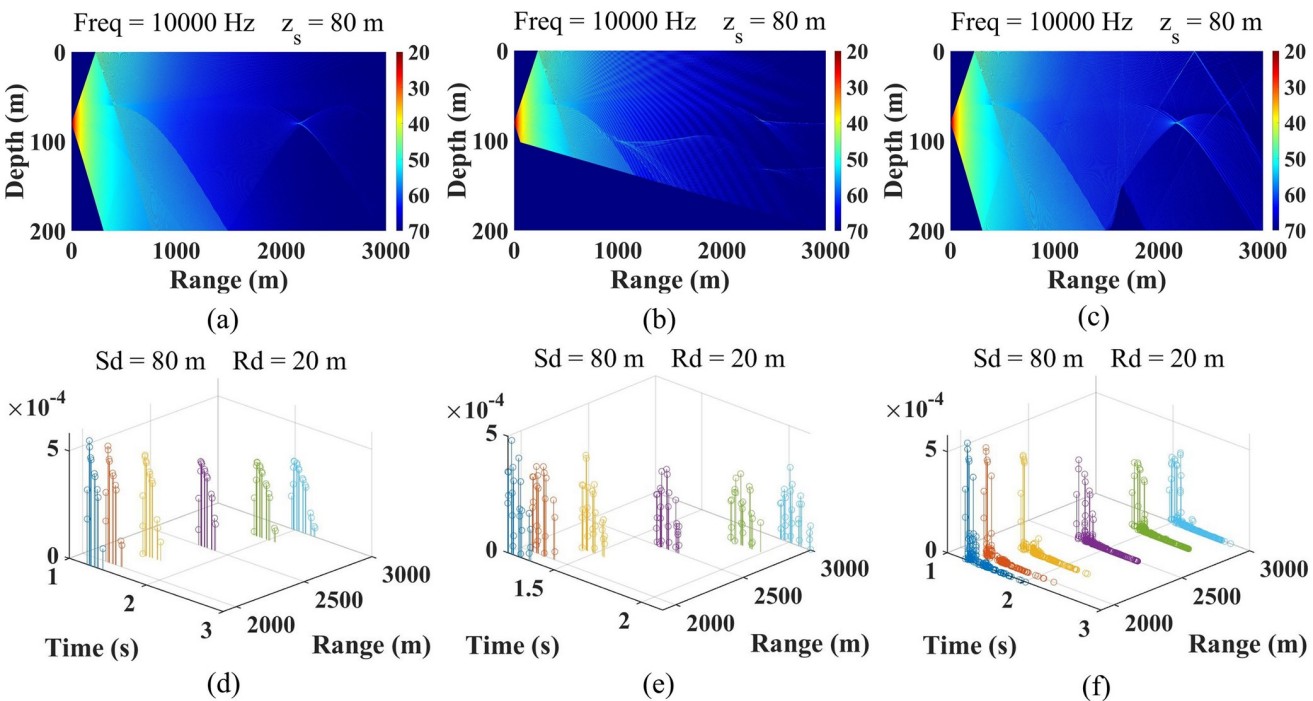

Fig 11. The propagation loss and channel impulse response of each receiving point in (a) marine environment 1, (b) marine environment 2 and (c) marine environment 3.

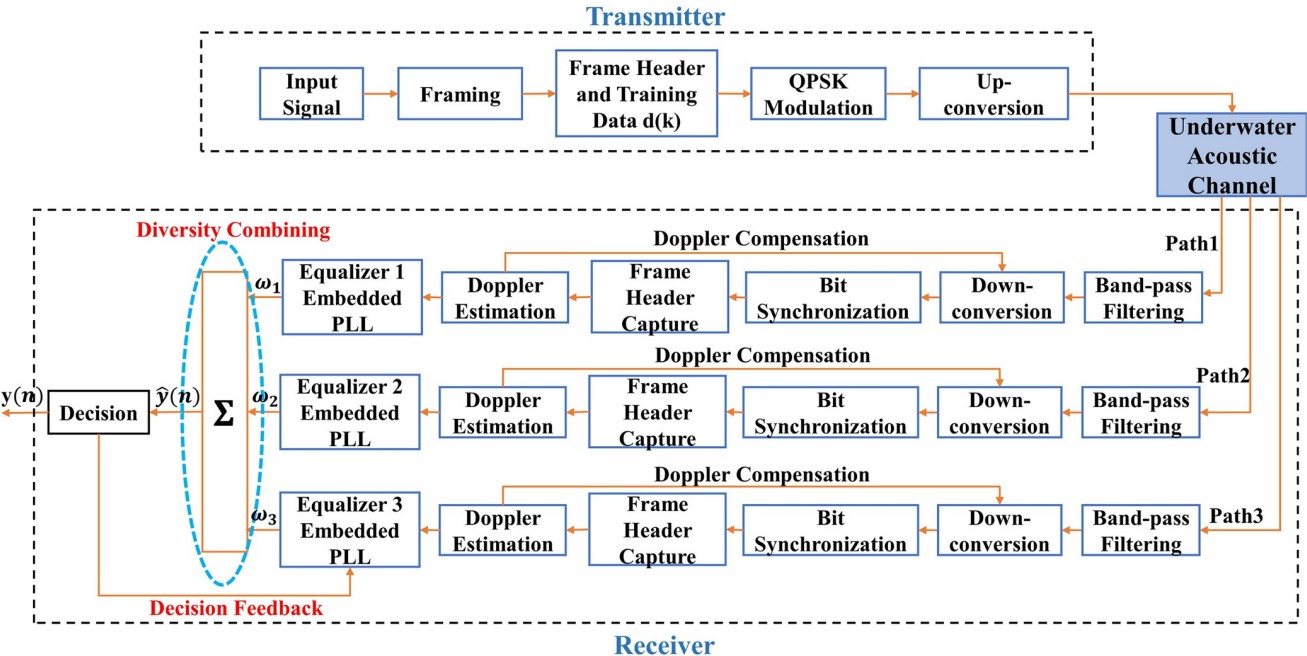

**Fig 12. The signal processing flow of the uplink.**

In the simulation, the Bellhop underwater acoustic channel model that we adopted is currently recognized as the theoretical model of the underwater acoustic channel, which is essentially a time-invariant channel model. Due to the time-invariant characteristics, the equalizer at the receiving end does not need to adjust the tap coefficient according to the channel change continuously, nor does it cause the sizeable mean square error of the output signal. So the diversity gain effect of the system is ideal. The actual marine underwater acoustic channel is complex and changeable, which will cause the communication system's decline in synchronization performance, the error of frame head and symbol synchronization, and the loss of lock and divergence of PLL of the equalizer, resulting in the increasing of bit error rate. In the actual underwater acoustic environment, we have verified the system's performance in the real lake experiment in the next chapter.

**Case 2**: Verify the impact of differe nt marine environments on the proposed system, and obtain the minimum number of diversity required for the proposed system in this paper and in [22, 23] to maintain communication in various marine environments. Assum that the system can communicate properly when the BER reaches $10^{-2}$. The underwater acoustic channel environment is set to 'Marine Environment 1', 'Marine Environment 2' and 'Marine Environment 3' respectively. The simulation results are shown in Fig 14.

In order to provide a clearer description of the data in the figures, we have compiled the data into Table 5.

As seen from Fig 14, with the gradual deterioration of the marine environment, the BER of the same diversity also gradually increases. The minimum diversity order required for the system to maintain communication is shown in Table 5. We can observe that in the cases of marine environment 1 and 3, the comparison algorithm requires a higher diversity order than DFE-PLL-MSIRC, thereby validating that DFE-PLL-MSIRC is more efficient than the comparison algorithm.

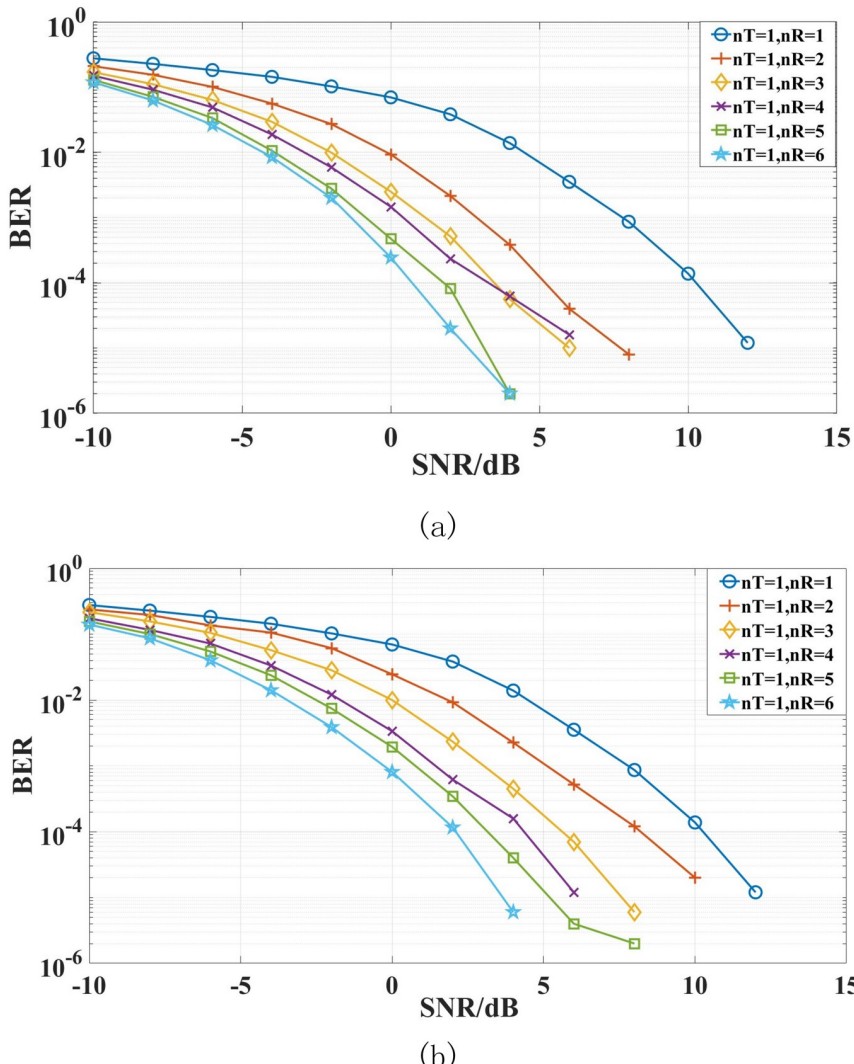

(a)

(b)

**Fig 13. The bit error rate (BER) of (a) DFE-PLL-MSIRC and (b) Comparison Algorithm [22, 23] (nT is the transmitter's number, nR is the receiver's number).**

**Case 3**: This simulation is to verify the influence of the transmit antennas' number on the system performance in the downlink under the same environmental parameters. The complex orthogonal transmission matrixes of two transmitting antennas and three transmitting antennas are designed as Eqs (32) and (33). The underwater acoustic channel environment is set to 'Marine Environment 1'. The simulation result is shown in Fig 15:

**Table 3. The required SNR (dB) of the system when BER = $10^{-4}$.**

| Algorithm | nT = 1<br>nR = 1 | nT = 1<br>nR = 2 | nT = 1<br>nR = 3 | nT = 1<br>nR = 4 | nT = 1<br>nR = 5 | nT = 1<br>nR = 6 |
|---|---|---|---|---|---|---|
| **DFE-PLL-MSIRC** | 10.5 | 5.3 | 3.4 | 3.1 | 2.2 | 0.7 |
| **Comparison Algorithm [22, 23]** | 10.5 | 8.2 | 5.4 | 4.4 | 3.1 | 2.1 |

**Table 4. The system's BER when SNR = 0.**

| Algorithm | nT = 1 nR = 1 | nT = 1 nR = 2 | nT = 1 nR = 3 | nT = 1 nR = 4 | nT = 1 nR = 5 | nT = 1 nR = 6 |
|---|---|---|---|---|---|---|
| DFE-PLL-MSIRC | 0.0697 | $9.302e^{-3}$ | $2.492e^{-3}$ | $1.464e^{-3}$ | $4.74e^{-4}$ | $2.44e^{-4}$ |
| Comparison Algorithm [22, 23] | 0.0697 | 0.02454 | $9.99e^{-3}$ | $3.356e^{-3}$ | $1.942e^{-3}$ | $8.08e^{-4}$ |

We can see from the above figure that COSTBC can obtain a 6 dB diversity gain compared with the point-to-point communication link in the underwater acoustic channel. In the environmental SNR of 5 dB, the BER of the two-transmitter and one-receiver model is 1.6 orders of magnitude lower than that of the point-to-point link. The BER of the three-transmitter and one-receiver model is two orders of magnitude lower than that of the point-to-point link, which improves the reliability and anti-fading of the communication system.

## Lake experiments and results

### Experiment environment

The lake experiment was conducted in Huating Lake, Anqing City, Anhui Province, in July 2021. The lake area is in good condition with slight waves.

Before the lake experiments, the hydrophones, and other experimental equipment have been professionally calibrated by the manufacturer. In the meantime, we have completed tasks such as testing electrical connections and ensuring communication link availability in a water tank environment. The transmitter sends a QPSK modulation signal through the transducer connected to the underwater acoustic communication apparatus module. The receiver collects underwater acoustic signals through a transducer connected to a digital collector. The obtained underwater acoustic signal is processed by Matlab. If the data can be successfully demodulated, it is proved that the communication link is passable and the electrical connection is correct.

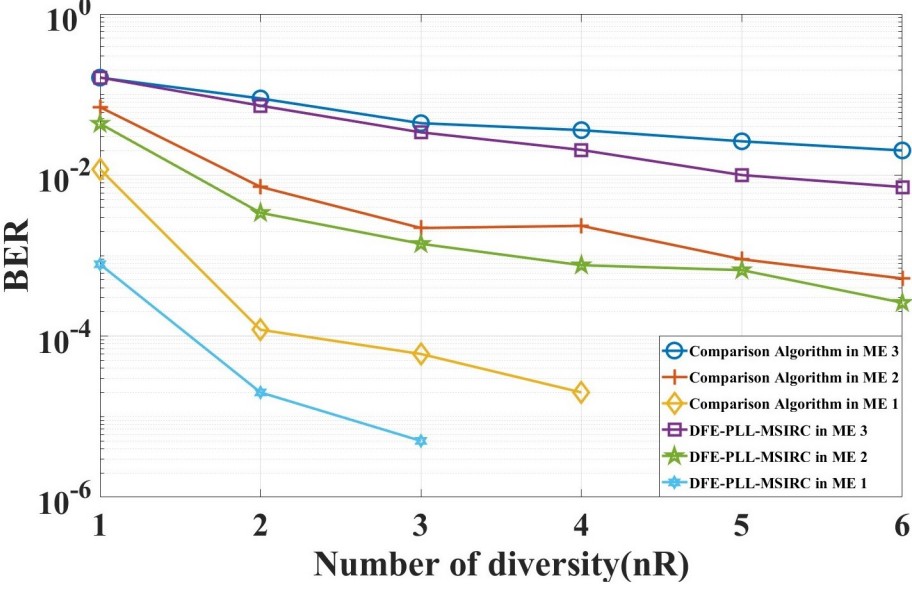

**Fig 14. BER of systems with different diversity in different marine environments (ME).**

**Table 5. The minimum number of diversity required for the system to maintain communication in various marine environments.**

| Algorithm | Marine Environment 1 | Marine Environment 2 | Marine Environment 3 |
|---|---|---|---|
| DFE-PLL-MSIRC | 1 | 2 | 6 |
| Comparison Algorithm [22, 23] | 2 | 2 | >6 |

The signal-transmitting ship goes to the intended test location, and the dock deploys three fixed diversity receivers. The distance between the transmitter and receivers is kept at 500 m and 1000 m respectively as shown in Fig 16. The equipment connection and layout of each ship are shown in Table 6. On the signal transmitting ship, the personal computer (PC) is connected to the underwater acoustic communication apparatus, and then the underwater acoustic communication apparatus is connected to the transducer. On the signal-receiving ship, the PC is connected to the digital collector, and then the digital collector is connected to the transducer. Among them, YOKOGAWA SL1000, HIOKI MR6000, and Altai represent the models of digital collectors. The QPSK modulated signal is processed by a power amplifier and sent through a transducer. All the diversity receivers synchronously collect multiple cycles of underwater acoustic signals through the digital collector connected to the transducer.

### Experiment results

The structures of the three channels of the transmitter and three receivers are collected and shown in Fig 17, which have apparent differences, and the correlation is not strong. The

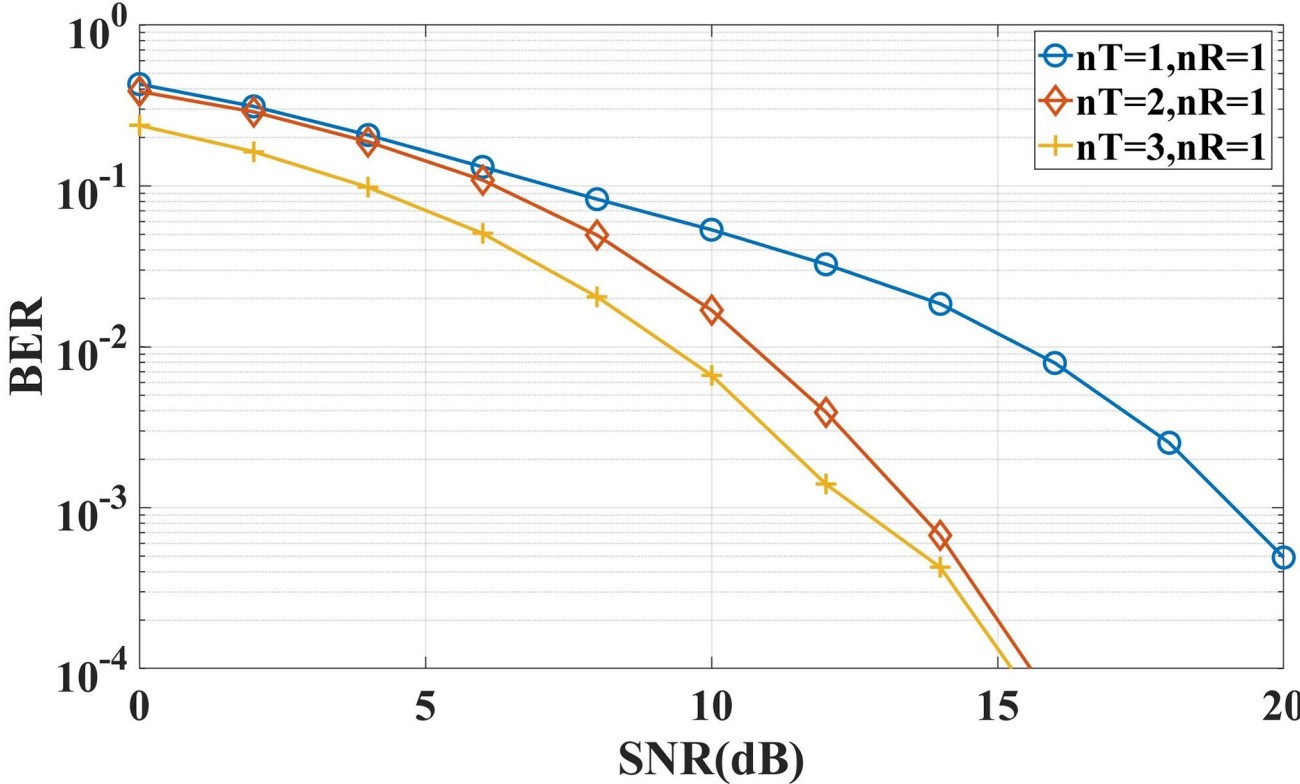

**Fig 15. BER of the system with different number of transmit antennas.**

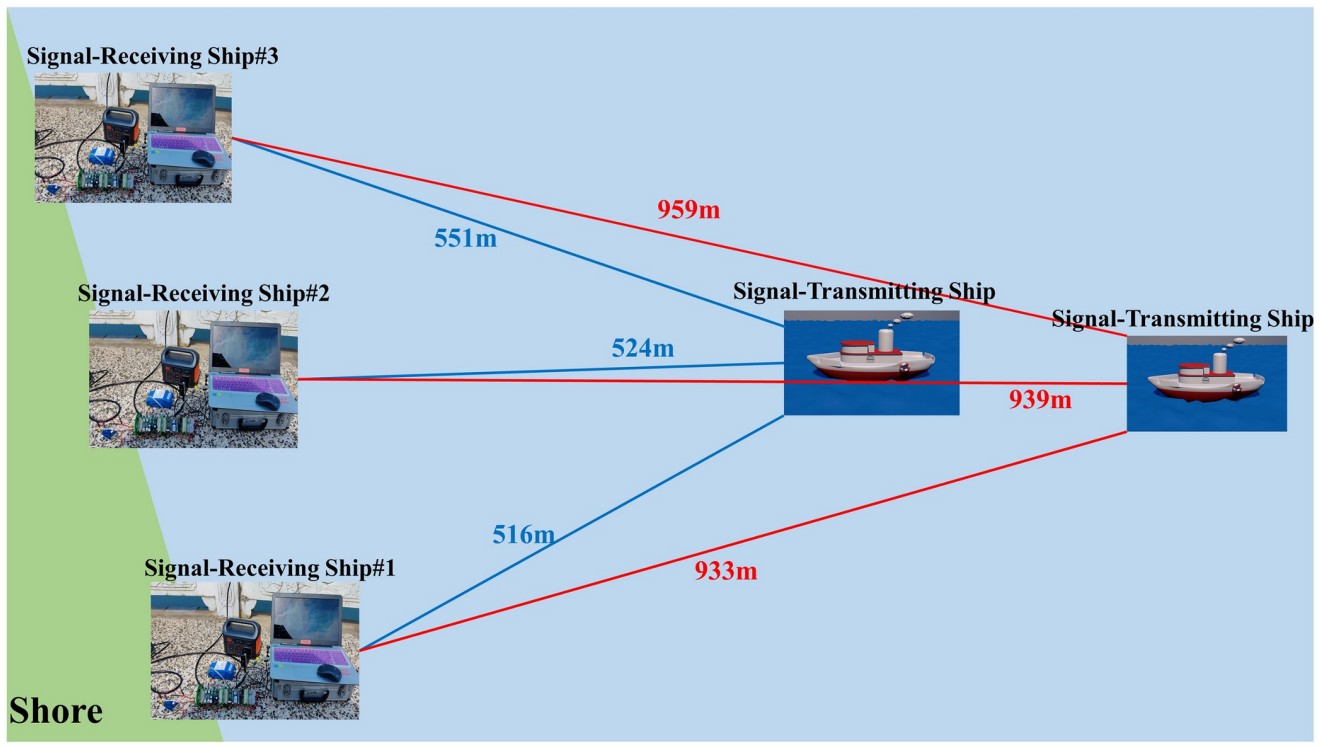

**Fig 16. The layout diagram of transmitter and receivers.**

distributed data are collected at three receiving points, and combined by DFE-PLL-MSIRC and comparison algorithm. The experimental results are shown in Figs 18 and 19.

It can be seen from Figs 18 and 19 that the proposed system's constellation diagrams are more convergent with the increase of the diversity number. The constellation map of DFE-PLL-MSIRC tends to approach the constellation points {1, 1}, {1, −1}, {−1, 1}, and {−1, −1} of the original QPSK signal, more so than the constellation map of the comparison algorithm. It can be seen from Table 7 that in the case of the transceiver spacing of 500 m, compared with point-to-point communication, the BER of two-point diversity combination can be reduced by an order of magnitude, and the BER of three-point diversity combination can be reduced to 0. In the case of the transceiver spacing of 1000 m, compared with point-to-point communication, the BER of two-point diversity combination can be reduced by 0.5 orders of magnitude, and the BER of three-point diversity combination can be reduced by 1 order of magnitude. Regardless of whether the transmission distance is 500 m or 1000 m, the BER of DFE-PLL-MSIRC is lower than that of the comparison algorithm. Because the actual

**Table 6. The equipment connection and layout.**

| Project | No. | Equipment Connection | Parameter | Layout |
|---|---|---|---|---|
| **Signal transmitter** | Ship #0 | PC-underwater acoustic communication apparatus-transducer | The baud rate of PC: 57600 B | The transducer is placed 10 m underwater by cable |
| **Signal receiver** | Ship #1<br>Ship #2<br>Ship #3 | PC-YOKOGAWA SL1000-transducer<br>PC-HIOKI MR6000-transducer<br>PC-Altai-transducer | The sampling rate of the collector: 500 ksps, no filter, channel 1 | The transducer is placed 10 m underwater by cable |

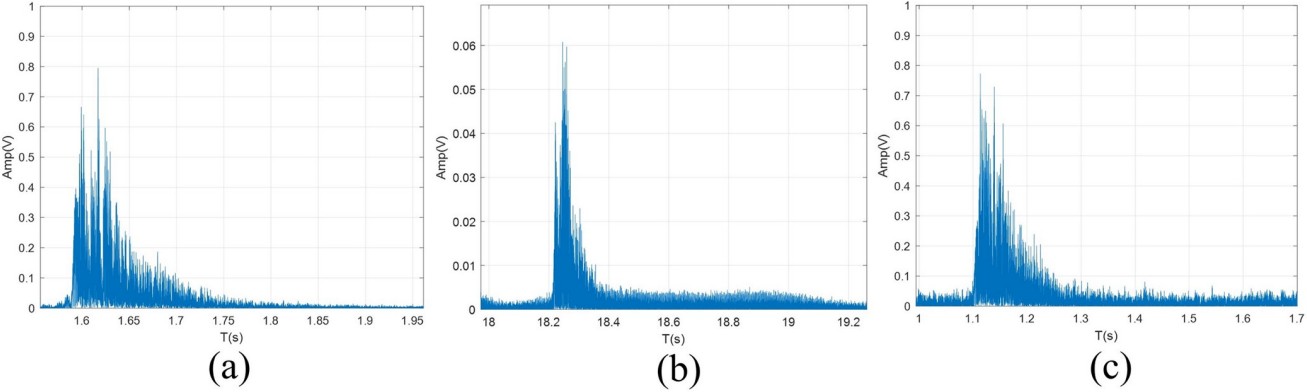

**Fig 17. The structures of (a) channel 1, (b) channel 2 and (c) channel 3.**

**Fig 18. The constellation diagrams of DFE-PLL-MSIRC (a) nT = 1, nR = 1 at 500 m, (b) nT = 1, nR = 2 at 500 m, (c) nT = 1, nR = 3 at 500 m, (d) nT = 1, nR = 1 at 1000 m, (e) nT = 1, nR = 2 at 1000 m and (f) nT = 1, nR = 3 at 1000 m.**

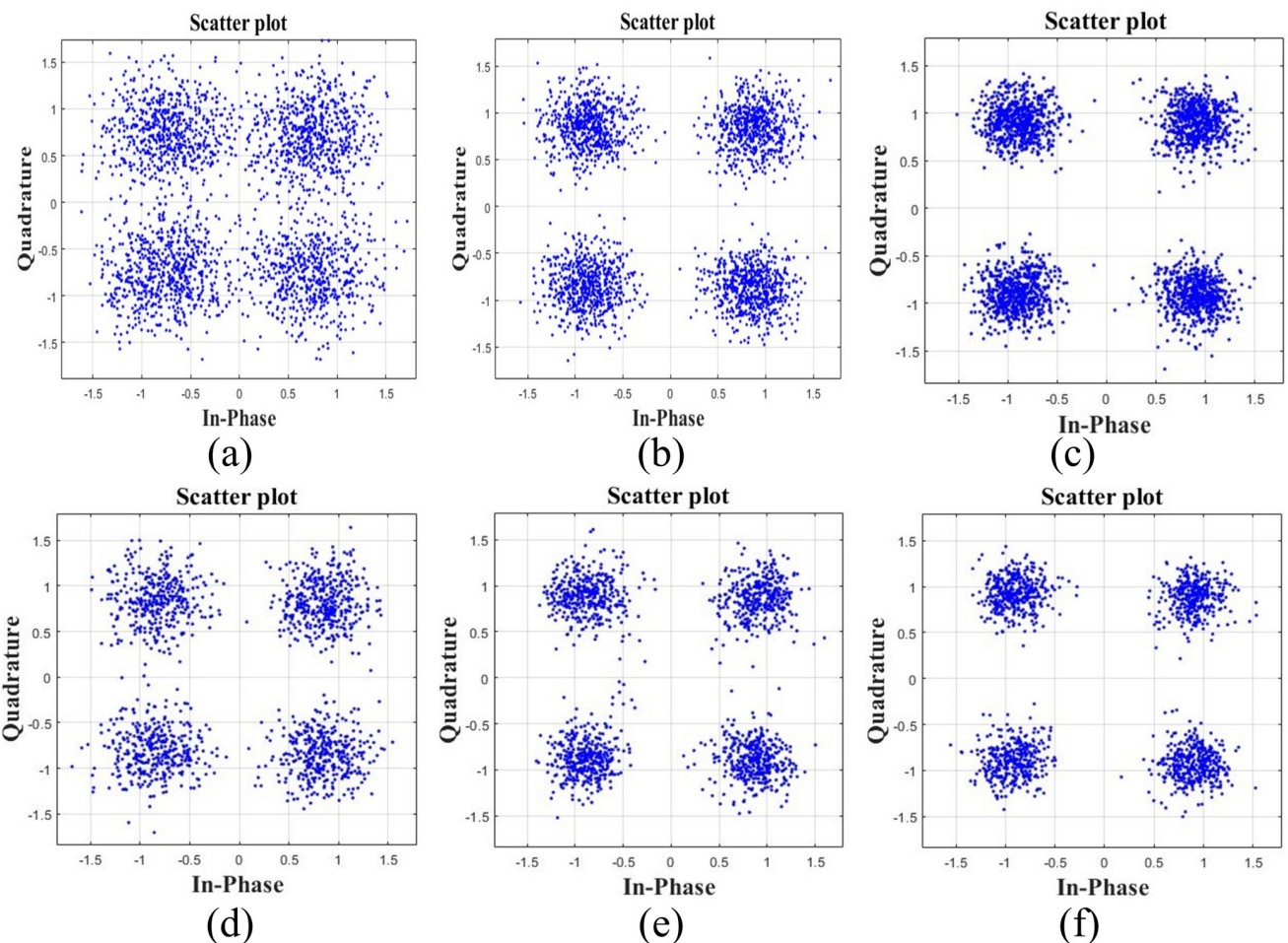

**Fig 19. The constellation diagrams of comparison algorithm (a) nT = 1, nR = 1 at 500 m, (b) nT = 1, nR = 2 at 500 m, (c) nT = 1, nR = 3 at 500 m, (d) nT = 1, nR = 1 at 1000 m, (e) nT = 1, nR = 2 at 1000 m and (f) nT = 1, nR = 3 at 1000 m.**

underwater acoustic channel is complex and changeable, which will cause the error of frame head and symbol synchronization, and the loss of lock and divergence of PLL of the equalizer, resulting in the increase of BER. However, compared with point-to-point communication, the diversity gain of the system is still apparent. The experimental results are generally consistent with the simulation results.

## Conclusion

This paper proposes DUA-DCS based on distributed buoys network. The uplink of the system adopts the receiving diversity processing mechanism of DFE-PLL-MSIR. Simulation and lake experiments results show that, compared to the point-to-point communication link system, DFE-PLL-MSIRC can achieve a diversity gain of more than 5.2 dB and obtain about 3 dB diversity gain more than the comparison algorithm. Moreover, the BER of DFE-PLL-MSIRC can be reduced by at least one order of magnitude, which is lower by at least 0.6 order of magnitude compared to the comparison algorithm. Considering the performance, the complexity of the system, and the cost of practical engineering, we propose to set the number of diversity

**Table 7. The BER of different diversity number at 500 m and 1000 m distance between transmitter and receivers.**

| Distance (m) | Frame | nT = 1, nR = 1 | | nT = 1, nR = 2 | | nT = 1, nR = 3 | |
|---|---|---|---|---|---|---|---|
| | | DFE-PLL-MSIRC | Comparison Algorithm | DFE-PLL-MSIRC | Comparison Algorithm | DFE-PLL-MSIRC | Comparison Algorithm |
| 500 | 1 | 0 | $1.6e^{-4}$ | 0 | $3.7e^{-5}$ | 0 | 0 |
| | 2 | $3.5e^{-3}$ | $7.5e^{-3}$ | 0 | $2.3e^{-4}$ | 0 | 0 |
| | 3 | $2.2e^{-3}$ | $6.0e^{-3}$ | 0 | 0 | 0 | 0 |
| | 4 | $2.16e^{-3}$ | $3.9e-3$ | $1.9e^{-4}$ | $4.4e^{-4}$ | 0 | $1.2e^{-6}$ |
| | 5 | $3.8e^{-3}$ | $8.7e-3$ | $1.3e^{-3}$ | $5.1e-3$ | 0 | $2.1e^{-5}$ |
| | 6 | $5.9e^{-3}$ | 0.0102 | $2.1e^{-3}$ | $6.5e^{-3}$ | 0 | $4.9e^{-5}$ |
| | Average | $2.927e^{-3}$ | $6.076e^{-3}$ | $5.98e^{-4}$ | $2.107e^{-3}$ | 0 | $1.187\ e^{-5}$ |
| 1000 | 1 | 0.0837 | 0.132 | 0.02 | 0.053 | 0.0026 | 0.0059 |
| | 2 | 0.0511 | 0.0947 | 0.0028 | 0.0061 | 0.00083 | 0.0016 |
| | 3 | 0.0206 | 0.056 | 0.0134 | 0.033 | 0.0012 | 0.0037 |
| | 4 | 0.0081 | 0.012 | 0.0049 | 0.0078 | 0.0021 | 0.0043 |
| | 5 | 0.0128 | 0.0453 | 0.0058 | 0.0098 | 0.0014 | 0.004 |
| | 6 | 0.0085 | 0.0124 | 0.0021 | 0.0054 | 0 | $7.33e^{-4}$ |
| | Average | $3.08e^{-2}$ | $5.87e^{-2}$ | $8.17e^{-3}$ | $1.918e^{-2}$ | $2.6e^{-3}$ | $3.37e^{-3}$ |

of DUA-DCS to 3 in practical engineering. The downlink of the system adopts the transmitting diversity processing mechanism of CSTBC. A new complex orthogonal transfer matrix is designed in this paper. The system can achieve a diversity gain compared with single-link communication. The construction of the system improves the reliability of information transmission between underwater users and shore stations, and realizes the interconnection between on-water and underwater. Further research on the distributed collaborative processing mechanism of underwater acoustic signals for spatial information networks, effectively extending the coverage of spatial information networks to the maritime domain, and achieving global stereo coverage of the space-sky-ground-sea integrated space information network is our future trend and direction.

## Supporting information

**S1 File. All data files are available from the figshare database.** (https://doi.org/10.6084/m9.figshare.23995809.v2).
(ZIP)

**S1 Appendix. Comparison table of the related works.**
(PDF)

## Author Contributions

**Conceptualization:** Yong Gao.

**Data curation:** Manli Zhou, Yong Gao.

**Formal analysis:** Manli Zhou.

**Funding acquisition:** Hao Zhang.

**Investigation:** Yingying Duan.

**Methodology:** Manli Zhou, Hao Zhang.

**Project administration:** Hao Zhang, Tingting Lv.

**Resources:** Hao Zhang.

**Software:** Manli Zhou.

**Supervision:** Hao Zhang, Tingting Lv.

**Validation:** Manli Zhou.

**Visualization:** Yingying Duan.

**Writing – original draft:** Manli Zhou.

**Writing – review & editing:** Manli Zhou.

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
