## [Decision Letter · Decision Letter 0]

7 Feb 2023

PONE-D-22-31101Distributed underwater acoustic diversity processing mechanism based on cross-media buoys networkPLOS ONE

Dear Dr. Zhou,

Thank you for submitting your manuscript to PLOS ONE. After careful consideration, we feel that it has merit but does not fully meet PLOS ONE’s publication criteria as it currently stands. Therefore, we invite you to submit a revised version of the manuscript that addresses the points raised during the review process.

We look forward to receiving your revised manuscript.

Kind regards,

Viacheslav Kovtun, Dr.Sc., Ph.D.

Academic Editor

PLOS ONE

Journal Requirements:

Reviewers' comments:

Reviewer's Responses to Questions

**Comments to the Author**

1. Is the manuscript technically sound, and do the data support the conclusions?

Reviewer #1: Partly

2. Has the statistical analysis been performed appropriately and rigorously? 

Reviewer #1: Yes

3. Have the authors made all data underlying the findings in their manuscript fully available?

Reviewer #1: No

4. Is the manuscript presented in an intelligible fashion and written in standard English?

Reviewer #1: Yes

5. Review Comments to the Author

Reviewer #1: 20230121

Manuscript Number: PONE-D-22-31101

Title: Distributed underwater acoustic diversity processing mechanism based on cross-media

buoys network

Authors: Manli Zhou, Hao Zhang, Tingting Lv, Yong Gao, Yingying Duan

Report:

The paper addresses the problem of single-link underwater acoustic communication. The single link

does not achieve the best and reliable results due to multiple reasons: the frequency selective fading,

multipath effect, and significant transmission loss. To overcome this problem,

the authors proposed a DUA-DCS approach to form a distributed non-coherent parallel communication link

and combine underwater acoustic signals at the waveform level to achieve the the best efficiency in information

transmission.

The paper is in general well-written with only a few errors. The main issues with this work are:

- The way of redacting the sentence: "... systems [9]. [10]conducted a ..." in page 2/15, first paragraph

is not very fortunate. It is suggested to improve the redaction and insert a space to separate words

while typing. For example as can be seen in: "... [17] design rate ..." in the second paragraph on page 2/15.

- The way of redacting the sentence: "... channels.In practice, ..." in page 2/15, first paragraph

is not very fortunate. It is suggested to improve the redaction and insert a space to

separate words after a period.

- It is suggested to improve the same in: "... [13]proposed a ..."; "... [14]studies a ..."; "... [16]cast space–time codes ...";

"... [19]proposed ..."; "... advantages:The phase ..."; "... compensation.The combining ..."

- In page 3/15, section "BELLHOP Gaussian beam tracking model", first paragraph, the redaction

is not very fortunate: "... sound field research22 ...". It is suggested to improve the redaction.

- It is suggested to improve the redaction in: "... model [24]is shown..." in page 3/15,

section "BELLHOP Gaussian beam tracking model", second paragraph.

- In page 4/15, it is not clear if conceptually the authors are referring to h(t) as the impulse response function,

defined in the equation (1), but at the same time, as h(t) to the transfer function. This could be improved.

- It is suggested to improve the redaction in page 4/15: "... based on a cross-media buoys network. Based on the ...".

In the same line, "based on" is used two times.

- In equation (4), page 6/15, the parameter "e" is not defined. It is suggested to improve the redaction.

- It is suggested to improve the redaction in: "... Initialization:ω (0) = ..."; "... Among them,y (n) = ..." in page 7/15,

and insert spaces to separate words while typing.

- In page 7/15, "... yb (n) = [y (n − 1), . . . , y (n − N3)]. ...", the parameter "N3" is not defined, or the question is if

it corresponds to N2, the order of FBF. This could be improved.

- In page 8/15, the style of the equation (16) is not very lucky: "... hj,i (t) = hj,i (t + T) = hj,i (t),t = 1, 2, ..., p".

This one could be improved.

- In page 8/15, the style of the equation (18) is not very lucky: "... xi = argmind2...". This one could be improved.

- In page 11/15, it is suggested to improve the redaction both in: "... Case 2:Verify ... ", and "... Case 3:This ..."

Insert spaces to separate words while typing.

- For consistency in writing, to refer to the equations, the authors use on page 8/15:

"... It satisfies Eq (12)...". However, on page 11/15, the authors employ: ".. .as Eq 14 and Eq 15. ...".

It is suggested to improve the redaction to be consistent in writing.

- In page 12/15, section Lake experiments and results. It would be opportune to know if the authors before

the field measurements have carried out, or not, a calibration procedure of the hydrophones and of all the experimental setup.

If some calibration procedure was carried out, it would be appropriated that the authors describe it.

- In page 12/15, it is suggested to improve the redaction in: "... Fig 15.The equipment ..." and insert

space to separate words while typing.

- Table 3, in page 12/15: The way of presenting the results in Table 3 is not very fortunate.

It is suggested to improve the presentation.

- Throughout the manuscript, each of the "Fig" abbreviations should carry a period. This is, for example, "Fig. 15".

This one could be improved.

- In the references, the authors only show one article published in 2019. No article is reviewed between

the years 2020 to date. This could be improved. It is suggested that the authors update the references

according to the state of the art.

6. PLOS authors have the option to publish the peer review history of their article (what does this mean?). If published, this will include your full peer review and any attached files.

Reviewer #1: **Yes: **Victor Poblete

---

## [Author Response · Author response to Decision Letter 0]

9 Mar 2023

Original Manuscript ID: PONE-D-22-31101 

Original Article Title: “Distributed underwater acoustic diversity processing mechanism based on cross-media buoys network”

To: PLOS ONE Editor

Re: Response to reviewers

Dear Editor,

Thank you for allowing a resubmission of our manuscript, with an opportunity to address the reviewers’ comments.

We are uploading (a) our point-by-point response to the comments (below) (Response to Reviewers), (b) an updated manuscript with yellow highlighting indicating changes (Revised Manuscript with Track Changes), and (c) a clean updated manuscript without highlights (Manuscript).

Best regards,

<Manli Zhou> et al.

Reviewer#1, Concern # 1: The way of redacting the sentence: "... systems [9]. [10]conducted a ..." in page 2/15, first paragraph is not very fortunate. It is suggested to improve the redaction and insert a space to separate words while typing. For example as can be seen in: "... [17] design rate ..." in the second paragraph on page 2/15.

Author response: We are sorry for this editing error. We have improved the redaction and inserted a space to separate words while typing. 

On page 2/17, first paragraph: "... systems [9]. [10] conducted a ...".

Reviewer#1, Concern # 2: The way of redacting the sentence: "... channels.In practice, ..." in page 2/15, first paragraph is not very fortunate. It is suggested to improve the redaction and insert a space to separate words after a period.

Author response: We are sorry for this editing error. We have improved the redaction and inserted a space to separate words after a period. 

On page 2/17, first paragraph: "... channels. In practice, ...".

Reviewer#1, Concern # 3: It is suggested to improve the same in: "... [13]proposed a ..."; "... [14]studies a ..."; "... [16]cast space–time codes ..."; "... [19]proposed ..."; "... advantages:The phase ..."; "... compensation.The combining ...".

Author response: We are sorry for this editing error. We have improved the redaction and inserted a space to separate words while typing. Since we have added new references, the numbers of references have also been updated.

On page 2/17, first paragraph: "To address this problem, [13] proposed a ..."; "... [14] studies a ...".

On page 2/17, second paragraph: "... [18] cast space–time codes ..."; "... The proposed transmission scheme [20] combines the ..."; "... [21] proposed ...".

On page 3/17: "... advantages: The phase ..."; "... compensation. The combining ...".

Reviewer#1, Concern # 4: In page 3/15, section "BELLHOP Gaussian beam tracking model", first paragraph, the redaction is not very fortunate: "... sound field research22 ...". It is suggested to improve the redaction.

Author response: We are sorry for this editing error. We have improved the redaction of Reference 22. Since we have added new references, the numbers of references have also been updated.

On page 4/17, first paragraph: "... sound field research [27] ...".

Reviewer#1, Concern # 5: It is suggested to improve the redaction in: "... model [24]is shown..." in page 3/15, section "BELLHOP Gaussian beam tracking model", second paragraph.

Author response: We are sorry for this editing error. We have improved the redaction and inserted a space to separate words while typing. Since we have added new references, the numbers of references have also been updated.

On page 4/17, second paragraph: "... model [29] is shown ...".

Reviewer#1, Concern # 6: In page 4/15, it is not clear if conceptually the authors are referring to h(t) as the impulse response function, defined in the equation (1), but at the same time, as h(t) to the transfer function. This could be improved.

Author response: Thank you for your valuable suggestions. I'm sorry for the conceptual confusion. If the underwater acoustic channel is regarded as a system, the input signal passes through the underwater acoustic channel to obtain the output signal. In the time domain, the impulse response is used to describe the characteristics of the system. That is, the input signal is convoluted with the impulse response to obtain the output signal of the system. In the frequency domain, the transfer function is used to describe the characteristics of the system. That is, the input signal is multiplied by the transfer function of the system to obtain the output signal of the system. In this paper, we model the underwater acoustic channel in the time domain. Therefore, h(t) should be described more accurately by impulse response. 

We have corrected the transfer function h(t) to the impulse response h(t) on page 4/17, section " Underwater acoustic channel impulse response h(t)".

Reviewer#1, Concern # 7: It is suggested to improve the redaction in page 4/15: "... based on a cross-media buoys network. Based on the ...". In the same line, "based on" is used two times.

Author response: Thank you for your valuable suggestions. We have rewritten this part for better readability.

On page 5/17, first paragraph: "This paper constructs a DUA-DCS based on a cross-media distributed buoys network. Because the multiple underwater acoustic communication links distributed in a certain area have the characteristics of mutual statistical independence, the system uses the buoys network with motorized deployment to convert the ’point-to-point’ variable parameter channel into a ’point-to-multipoint’ or ’multi-point-to-point’ quasi-constant parameter stationary channel.".

Reviewer#1, Concern # 8: In equation (4), page 6/15, the parameter "e" is not defined. It is suggested to improve the redaction.

Author response: Thank you for your valuable suggestions. We have defined the parameter "e" and updated the manuscript.

On page 7/17, fourth paragraph: "e(i) denotes the error between the desired output value d(i) of the equalizer and the decision value y ^(i) at the time i."

Reviewer#1, Concern # 9: It is suggested to improve the redaction in: "... Initialization:ω (0) = ..."; "... Among them,y (n) = ..." in page 7/15, and insert spaces to separate words while typing.

Author response: We are sorry for this editing error. We have improved the redaction and inserted a space to separate words while typing.

On page 7/17: "... Initialization: ω (0) = ..."; "... Among them, y (n) = ...".

Reviewer#1, Concern # 10: In page 7/15, "... y_b (n)=[y(n-1),⋯,y(n-N_3 )]. ...", the parameter "N_3" is not defined, or the question is if it corresponds to "N_2", the order of FBF. This could be improved.

Author response: Thank you for your valuable suggestions. The parameter "N_3" corresponds to "N_2", that is, the order of FBF. We have corrected it in the manuscript.

On page 7/17: "y_b (n)=[y(n-1),⋯,y(n-N_2 )]".

Reviewer#1, Concern # 11: In page 8/15, the style of the equation (16) is not very lucky: "... h_(j,i) (t)=h_(j,i) (t+T)=h_(j,i),t=1,2,⋯p ". This one could be improved.

Author response: Thank you for your valuable suggestions. We have modified equation (16) according to the correct format. 

On page 10/17, equation (16): "h_(j,i) (t)=h_(j,i) (t+T)=h_(j,i) 

Where t=1,2,⋯p."

Reviewer#1, Concern # 12: In page 8/15, the style of the equation (18) is not very lucky: "... xi = argmind2...". This one could be improved.

Author response: Thank you for your valuable suggestions. We have modified equation (18) according to the correct format. 

On page 10/17, equation (18): " x ^_i=argmin┬(x ^_i ϵS)⁡(d^2 (x ~_i,x ^_i))+(∑_(i=1)^(N_i)∑_(j=1)^(N_t)|h_(j,i) |^2-1)|x_i |^2 

Where i=1,2,⋯k."

Reviewer#1, Concern # 13: In page 11/15, it is suggested to improve the redaction both in: "... Case 2:Verify ... ", and "... Case 3:This ..." Insert spaces to separate words while typing.

Author response: We are sorry for this editing error. We have improved the redaction and inserted a space to separate words while typing. 

On page 12/17, second paragraph: "... Case 2: Verify ... ".

On page 12/17, fourth paragraph: "... Case 3: This ...".

Reviewer#1, Concern # 14: For consistency in writing, to refer to the equations, the authors use on page 8/15: "... It satisfies Eq (12)...". However, on page 11/15, the authors employ: "...as Eq 14 and Eq 15. ...". It is suggested to improve the redaction to be consistent in writing.

Author response: Thank you for your valuable suggestions. For consistency in writing, we have corrected "...as Eq 14 and Eq 15 ..." to be consistent with "... It satisfies Eq (12) ...".

On page 12/17, fourth paragraph: " Case 3: This simulation is to verify the influence of the transmit antennas’ number on the system performance in the downlink under the same environmental parameters. The complex orthogonal transmission matrixes of two transmitting antennas and three transmitting antennas are designed as Eq (14) and Eq (15)."

Reviewer#1, Concern # 15: In page 12/15, section Lake experiments and results. It would be opportune to know if the authors before the field measurements have carried out, or not, a calibration procedure of the hydrophones and of all the experimental setup. If some calibration procedure was carried out, it would be appropriated that the authors describe it.

Author response: Thank you for your valuable suggestions. Before the field measurements, we carried out a calibration procedure of the hydrophones and all the experimental setups. We updated the manuscript by describing this part in detail.

On page 12/17: Add "Before the lake experiments, the hydrophones, and other experimental equipment have been professionally calibrated by the manufacturer. In the meantime, we have completed tasks such as testing electrical connections and ensuring communication link availability in a water tank environment. The transmitter sends a QPSK modulation signal through the transducer connected to the underwater acoustic communication apparatus module. The receiver collects underwater acoustic signals through a transducer connected to a digital collector. The obtained underwater acoustic signal is processed by Matlab. If the data can be successfully demodulated, it is proved that the communication link is passable and the electrical connection is correct."

Reviewer#1, Concern # 16: In page 12/15, it is suggested to improve the redaction in: "... Fig 15.The equipment ..." and insert space to separate words while typing.

Author response: We are sorry for this editing error. We have improved the redaction and inserted a space to separate words while typing. 

On page 13/17, first paragraph: "... Fig. 15. The equipment ...".

Reviewer#1, Concern # 17: Table 3, in page 12/15: The way of presenting the results in Table 3 is not very fortunate. It is suggested to improve the presentation.

Author response: Thank you for your valuable suggestions. We modified Table 3 according to the standard of the three-line table. To make readers better understand, Table 3 is explained in detail in the manuscript. At the same time, to maintain the consistency of the format of all tables, we also modified Table 1, Table 2, and Table 4 to the format of the three-line table.

On page 13/17, first paragraph: add "On the signal transmitting ship, the personal computer (PC) is connected to the underwater acoustic communication apparatus, and then the underwater acoustic communication apparatus is connected to the transducer. On the signal-receiving ship, the PC is connected to the digital collector, and then the digital collector is connected to the transducer. Among them, YOKOGAWA SL1000, HIOKI MR6000, and Altai represent the models of digital collectors."

Reviewer#1, Concern # 18: Throughout the manuscript, each of the "Fig" abbreviations should carry a period. This is, for example, "Fig. 15". This one could be improved.

Author response: Thank you very much for your valuable comments, we have modified each of the "Fig" to "Fig." according to your comments.

Reviewer#1, Concern # 19: In the references, the authors only show one article published in 2019. No article is reviewed between the years 2020 to date. This could be improved. It is suggested that the authors update the references according to the state of the art.

Author response: Thank you for your valuable suggestion. We updated the manuscript by adding new references between the years 2020 to date.

On page 2/17, first paragraph: add [15] in " In [15], the received signal is split into two streams. One stream is equalized after time reversal (TR) combining. Another stream is flipped in the time domain, processed by TR-combining and equalization, and then flipped in the time domain again. After processing, each stream is conducted diversity combining."

On page 2/17, first paragraph: add [16] in " [16] employed spatial diversity and antenna beamforming methods to significantly improve the signal quality performance of wireless networks and enhance the reliability by decreasing the bit error probability."

On page 3/17, first paragraph: add [24] in " In [24], joint time-reversal space-time block coding and adaptive equalization filtered multitone underwater acoustic communication method was proposed. The effectiveness of the proposed method is verified by simulation analysis and real experimental data collected from an indoor pool communication trial."

On page 3/17, first paragraph: add [25] in " [25] proposed a generalized space-time block coded spatial modulation scheme for open-loop massive multiple-input and multiple-output downlink communication systems. The information bits are divided into multiple groups with each group modulated by the spatial modulation, where the SM symbols are invoked for OSTBC and quasi-orthogonal STBC structures."

On page 3/17, first paragraph: add [26] in " [26] proposed space-time block coded non-orthogonal multiple access (STBC-NOMA) for underwater acoustic sensor networks (UASNs) to improve reliability by exploiting transmit diversity and spectral efficiency. Results show that STBC-NOMA can significantly enhance the performance of UASNs without the need for prior channel state information status at the transmitter."

[15] Kim H, Choi KH, Choi JW, Bae HS. Bidirectional Equalization for Long-Range Underwater Acoustic Communication in BLAC18. In: 2019 Eleventh International Conference on Ubiquitous and Future Networks (ICUFN). IEEE; 2019. p. 52–53.

[16] Mehta R. Optimal receive beamforming in spatial antenna diversity system using evolutionary genetic algorithm. Array. 2021; 10:100053.

[24] Sun L, Yan M, Li H, Xu Y. Joint Time-Reversal Space-Time Block Coding and Adaptive Equalization for Filtered Multitone Underwater Acoustic Communications. Sensors. 2020;20(2):379.

[25] Xiao L, Chen D, Hemadeh I, Xiao P, Jiang T. Generalized space-time block coded spatial modulation for open-loop massive MIMO downlink communication systems. IEEE Transactions on Communications. 2020;68(11):6858–6871.

[26] Goutham V, Harigovindan V. Space–time block coded non-orthogonal multiple access for performance enhancement of underwater acoustic sensor networks. ICT Express. 2022;8(1):117–123.

Thanks again for your advice and I hope to learn more from you.

---

## [Decision Letter · Decision Letter 1]

26 Jun 2023

PONE-D-22-31101R1Distributed underwater acoustic diversity processing mechanism based on cross-media buoys networkPLOS ONE

Dear Dr. Zhou,

Thank you for submitting your manuscript to PLOS ONE. After careful consideration, we feel that it has merit but does not fully meet PLOS ONE’s publication criteria as it currently stands. Therefore, we invite you to submit a revised version of the manuscript that addresses the points raised during the review process.

We look forward to receiving your revised manuscript.

Kind regards,

Ravikumar CV, Ph.D

Academic Editor

PLOS ONE

Additional Editor Comments:

Dear Authors, based on the reviewers' remarks, your paper requires significant modifications. Please get the track revisions version ready for the second round of approval.

Comments from PLOS Editorial Office:

We note that one or more reviewers has recommended that you cite specific previously published works. As always, we recommend that you please review and evaluate the requested works to determine whether they are relevant and should be cited. It is not a requirement to cite these works. We appreciate your attention to this request.

Reviewers' comments:

Reviewer's Responses to Questions

**Comments to the Author**

1. If the authors have adequately addressed your comments raised in a previous round of review and you feel that this manuscript is now acceptable for publication, you may indicate that here to bypass the “Comments to the Author” section, enter your conflict of interest statement in the “Confidential to Editor” section, and submit your "Accept" recommendation.

Reviewer #1: All comments have been addressed

Reviewer #2: All comments have been addressed

2. Is the manuscript technically sound, and do the data support the conclusions?

Reviewer #1: Yes

Reviewer #2: Partly

3. Has the statistical analysis been performed appropriately and rigorously? 

Reviewer #1: Yes

Reviewer #2: Yes

4. Have the authors made all data underlying the findings in their manuscript fully available?

Reviewer #1: Yes

Reviewer #2: Yes

5. Is the manuscript presented in an intelligible fashion and written in standard English?

Reviewer #1: Yes

Reviewer #2: Yes

6. Review Comments to the Author

Reviewer #1: 20230520

Manuscript Number: PONE-D-22-31101R1

Title: Distributed underwater acoustic diversity processing mechanism based on cross- media buoys network

Authors: Manli Zhou

Ocean University of China College of Information Science and Engineering Qingdao, CHINA

Report:

In my opinion, it can be seen that the authors carefully reviewed the article, following each of the suggestions and observations that were made. This final version of the article is a powerful contribution to the disciplinary field of underwater acoustic communication. My congratulations to each of the authors for this valuable work done.

Reviewer #2: In this article, “Distributed underwater acoustic diversity processing mechanism based on cross media buoys network”. The author must improve the manuscript. However, I have some comments and suggestions for the authors as follows:

1. It is better to revise or suggest another appropriate title for this Manuscript to reflect the details of the work.

2. The abstract does not communicate effectively; it should be changed.

3. The paper organization should be revised with more details about the proposed work in this manuscript.

4. In particular, the paper’s contribution is unclear compared to existing works in the literature, and there lacks a necessary comparison with the existing image enhancement methods. Thus, the current version cannot be recommended for acceptance.

5. Please add a list of Abbreviations (Acronym) after Conclusion Section,

6. In the introduction, the authors did not give clearly describe the new methods and contributions that lead to improving the comprehensive survey of image enhancement methods compared to existing works. Please explain the new comprehensive contributions of this work (methodology or algorithm?) that will improve the image enhancement existing works more specifically.

7. The authors mentioned only some related work, but without enough details. So it is necessary to add existing works in the related work section with the comparison table to improve this manuscript. Please revise it.

8. There are too many spelling and grammar mistakes in the paper. It needs proper spelling and grammar checking.

9. Most mentioned Figures includes small details; so they are not clear for the readers to distinguish the difference and the performance improvement of image enhancement methods. Add full description for all the results.

10. In conclusion, you can show briefly the percentage of improvement of image enhancement methods interms of quantitative evaluation measures.

11. What are the trends and future direction of applying those proposed method? Please give your suggestions.

12. Add some recently papers (2020-2023) in the manuscript.

https://www.mdpi.com/1424-8220/23/10/4844

https://doi.org/10.3390/electronics12061287

https://www.mdpi.com/2224-2708/11/4/64

https://www.hindawi.com/journals/jcnc/2022/9418392

https://doi.org/10.3390/jsan11040064

7. PLOS authors have the option to publish the peer review history of their article (what does this mean?). If published, this will include your full peer review and any attached files.

Reviewer #1: **Yes: **Dr. Víctor Poblete Ramírez

Reviewer #2: No

---

## [Author Response · Author response to Decision Letter 1]

22 Aug 2023

Original Manuscript ID: PONE-D-22-31101R1 

Original Article Title: “Distributed underwater acoustic diversity processing mechanism based on cross-media buoys network”

To: PLOS ONE Editor

Re: Response to reviewers

Dear Editor,

Thank you for allowing a resubmission of our manuscript, with an opportunity to address the reviewers’ comments.

We are uploading (a) our point-by-point response to the comments (below) (Response to Reviewers), (b) an updated manuscript with yellow highlighting indicating changes (Revised Manuscript with Track Changes), and (c) a clean updated manuscript without highlights (Manuscript).

Best regards,

<Manli Zhou> et al.

This part is the response to the second reviewer's comments in the second round of revisions.

Reviewer#2, Concern # 1: It is better to revise or suggest another appropriate title for this Manuscript to reflect the details of the work.

Author response: Thank you for your valuable suggestion. We have revised the title of the manuscript to “Spatial diversity processing mechanism based on the distributed underwater acoustic communication system”.

Reviewer#2, Concern # 2: The abstract does not communicate effectively; it should be changed.

Author response: Thank you for your valuable suggestion. We have revised the abstract to make it clearer and more readable.

On page 1, first paragraph: “To address the problem of unreliable single-link underwater acoustic communication caused by large signal delays and strong multipath effects in shallow water environments, this paper proposes a distributed underwater acoustic diversity communication system (DUA-DCS). DUA-DCS employs a maneuverable distributed cross-medium buoy network to form multiple distributed, non-coherent, and parallel communication links. In the uplink, a receiving diversity processing mechanism of joint decision feedback equalizer embedded phase-locked loop and maximum signal-to-interference ratio combining (DFE-PLL-MSIRC) is proposed to achieve waveform-level diversity combining of underwater signals. A phase-locked loop module is embedded in each branch of the decision feedback equalizer to eliminate the residual frequency and phase errors after Doppler compensation. Meanwhile, the combining coefficients are determined based on the maximum signal-to-interference ratio criterion, taking into account the residual inter-symbol interference after equalization, resulting in efficient and accurate computation. Additionally, the combined decision values are fed back to the feedback filters in each branch to ensure more accurate feedback output. Simulation and lake experiment results demonstrate that, compared to the single-link communication system, DFE-PLL-MSIRC can achieve a diversity gain of more than 5.2 dB and obtain about 3 dB more diversity gain than the comparison algorithm. And the BER of DFE-PLL-MSIRC can be reduced by at least one order of magnitude, which is lower by at least 0.6 order of magnitude compared to the comparison algorithm. In the downlink, a transmitting diversity processing mechanism of complex orthogonal space-time block coding (COSTBC) is proposed. By utilizing a newly designed generalized complex orthogonal transmission matrix, complete transmission diversity can be achieved at the coding rate of 3/4. Compared to the single-link communication system, the system can achieve a diversity gain of more than 6 dB.”

Reviewer#2, Concern # 3: The paper organization should be revised with more details about the proposed work in this manuscript.

Author response: Thank you for your valuable suggestion. In order to provide a more detailed description of the work proposed in the manuscript, we have revised the organization structure of the manuscript and included the following content:

On page 4 line 134-138, add: “The remainder of this manuscript is organized as follows. Section 2 introduces the underwater acoustic channel model. Section 3 introduces the system architecture. Section 4 introduces the spatial diversity processing mechanism. The results of the simulations and lake experiments are presented in Section 5 and 6, respectively. In Section 7, we finally conclude.”

On page 6 line 225-252, we have added the subsubsection “Decision feedback equalizer”.

On page 8 line 269-290, we have added the subsubsection “Phase-locked loop”.

On page 9 line 291-336, we have revised the subsubsection “DFE-PLL-MSIRC”.

Reviewer#2, Concern # 4: In particular, the paper’s contribution is unclear compared to existing works in the literature, and there lacks a necessary comparison with the existing image enhancement methods. Thus, the current version cannot be recommended for acceptance.

Author response: Thank you for your valuable suggestion. I 'm sorry to make you confused. The main theme of this paper is not about image enhancement. This paper's primary focus is constructing a distributed underwater acoustic diversity communication system to address the issue of unreliability in single-link underwater acoustic communication. We propose a decision feedback space receiving diversity equalizer embedded phase-locked loop in the uplink, and a transmitting diversity receiver for complex orthogonal space-time coding in the downlink.

We have made revisions to the introduction section and made clear descriptions of the contributions of this paper. Meanwhile, we have provided a comprehensive comparison with existing works in the literature on spatial diversity equalizers in underwater acoustics.

On page 2 line 42-57, add: “However, most communication technologies initially developed for terrestrial wired and wireless channels do not apply to the underwater acoustic environment [20]. In underwater acoustic communication, spatial diversity is also receiving significant attention as one of the techniques to improve performance under challenging channel conditions. For the joint reception of multiple signals in frequency-selective fading channels, antenna arrays and equalizers are used for diversity reception to improve the quality of wireless communication systems [21]. [22, 23] proposed multichannel DFE receiver for underwater acoustic communications. Received signals are processed by a bank of adaptive linear filters that jointly perform matched filtering and feed-forward equalization. Adaptive phase synchronization is then performed on each branch before the signals are combined and passed to a single DFE feedback and decision loop. In [24], the received signal is split into two streams. One stream is equalized after time reversal (TR) combining. Another stream is flipped in the time domain, processed by TR-combining and equalization, and then flipped in the time domain again. After processing, each stream is conducted diversity combining.”

On page 3 line 109-133, we have made revisions: “...The main contributions of this paper are as follows. 

1. The distributed... 2. To input different... 3. We propose a receiving... 4. We propose a transmitting...”.

On page 9 line 303-308, add: “Compared with the SIMO-DFE system described in [22, 23], this paper combines spatial diversity techniques with an adaptive decision feedback equalizer embedded PLL at the receiver. Phase-compensated and equalization are performed separately on each branch before diversity combining. Then the output of each equalizer is combined with the maximum signal-to-interference ratio criterion (MSIRC). The combined signal is decided and input into the FBF of each equalizer and the phase detector of each branch, rather than the combined signal itself.”

Reviewer#2, Concern # 5: Please add a list of Abbreviations (Acronym) after Conclusion Section.

Author response: Thank you for your valuable suggestion. We have added a list of Abbreviations after Conclusion Section on page 19.

Abbreviations 

DFE-PLL-MSIRC decision feedback equalization embedded phase-locked loop and maximum 

signal-to-interference ratio combining

DUA-DCS distributed underwater acoustic diversity communication system

COSTBC complex orthogonal space-time block code

DAS distributed antennas system

DFE decision feedback equalization

SNR signal-to-noise ratio

MRC maximal ratio combining

OFDM orthogonal frequency division multiplexing

TR time reversal

STBC space-time block code

OSTBC orthogonal space-time block code

GSTBC-SM generalized space-time block coded spatial modulation

MIMO multiple-input and multiple-output

SM spatial modulation

Q-OSTBC quasi-orthogonal space-time block code

STBC-NOMA space–time block coded non-orthogonal multiple access

UASNs underwater acoustic sensor networks

NSFC National Natural Science Foundation of China

PLL phase-locked loop

RF radio frequency

FFF feedforward filter

FBF feedback filter

RLS recursive least square

VCO voltage-controlled oscillator

SIMO single input multiple output

MSIRC maximum signal-to-interference ratio criterion

MSE mean square error

MISO multi-input single-output

BER bit error rate

Reviewer#2, Concern # 6: In the introduction, the authors did not give clearly describe the new methods and contributions that lead to improving the comprehensive survey of image enhancement methods compared to existing works. Please explain the new comprehensive contributions of this work (methodology or algorithm?) that will improve the image enhancement existing works more specifically.

Author response: Thank you for your valuable suggestion. In order to provide a clearer description of the novel algorithm and its contributions in this paper, we have made revisions to the introduction section. At the same time, I 'm sorry to make you confused. The main theme of this paper is not about image enhancement. We have added relevant references for spatial diversity equalizers for underwater acoustic communication. Additionally, we have provided a detailed description of the contributions of this paper.

On page 2 line 42-57, add: “However, most communication technologies initially developed for terrestrial wired and wireless channels do not apply to the underwater acoustic environment [20]. In underwater acoustic communication, spatial diversity is also receiving significant attention as one of the techniques to improve performance under challenging channel conditions. For the joint reception of multiple signals in frequency-selective fading channels, antenna arrays and equalizers are used for diversity reception to improve the quality of wireless communication systems [21]. [22, 23] proposed multichannel DFE receiver for underwater acoustic communications. Received signals are processed by a bank of adaptive linear filters that jointly perform matched filtering and feed-forward equalization. Adaptive phase synchronization is then performed on each branch before the signals are combined and passed to a single DFE feedback and decision loop. In [24], the received signal is split into two streams. One stream is equalized after time reversal (TR) combining. Another stream is flipped in the time domain, processed by TR-combining and equalization, and then flipped in the time domain again. After processing, each stream is conducted diversity combining.”

On page 3 line 109-133, the main contribution of the paper is revised as follows:

1. The distributed underwater acoustic diversity communication system (DUA-DCS) that we propose can solve the issue of unreliable point-to-point underwater acoustic communication links in shallow water environments and achieve a waveform-level diversity combination of underwater acoustic signals. 

2. To input different complex marine environment parameters into the bellhop model to obtain the underwater acoustic channel models under different shallow seabed topography and sediment environments and calculate the performance of the proposed system in these different marine environments. 

3. We propose a receiving diversity processing mechanism of joint decision feedback equalization embedded phase-locked loop and maximum signal-to-interference ratio combining (DFE-PLL-MSIRC), which exhibits the following advantages: 

• To eliminate the residual frequency and phase errors after Doppler compensation, a PLL module is embedded in each branch of the DFE.

• The combining coefficient of each branch adopts the maximum signal-to-interference ratio criterion, which considers the residual inter-symbol interference after equalization in addition to additive noise interference, leading to efficient and accurate computation. 

• The combined decision output is fed back to the feedback filters in each branch to ensure more accurate feedback output. 

•Considering system performance, complexity, and practical engineering costs, some engineering recommendations are provided.

4. We propose a transmitting diversity processing mechanism of complex orthogonal space-time block code (COSTBC). By utilizing a newly designed generalized complex orthogonal transmission matrix, complete transmission diversity can be achieved at the coding rate of 3/4.

Reviewer#2, Concern # 7: The authors mentioned only some related work, but without enough details. So it is necessary to add existing works in the related work section with the comparison table to improve this manuscript. Please revise it.

Author response: Thank you for your valuable suggestion. In order to provide a more detailed description of the related work mentioned in the manuscript, we have included the following content. Furthermore, we have added a comparison table of existing works in the appendix.

On page 1 line 2-8, add: “The breadth and depth of research on smart oceans and transparent oceans for ocean development and utilization have been continuously strengthened in recent years. By utilizing the features of broad coverage, fade resistance, and minimal required switching provided by the distributed antenna systems (DAS) [1, 2], underwater sensor nodes collect and transmit data through point-to-point and multipoint-to-multipoint communication to construct a distributed underwater sensor network for the effective transmission of information [3–6].”

On page 2 line 35-38, add: “In [18], the Maximum Ratio Combining (MRC) diversity algorithm is combined with the Orthogonal Frequency Division Multiplexing (OFDM) system to improve the received signal at the receiver. The primary idea behind MRC diversity is to improve the received signal at the receiver.”

On page 2 line 42-57, add: “However, most communication technologies initially developed for terrestrial wired and wireless channels do not apply to the underwater acoustic environment [20]. In underwater acoustic communication, spatial diversity is also receiving significant attention as one of the techniques to improve performance under challenging channel conditions. For the joint reception of multiple signals in frequency-selective fading channels, antenna arrays and equalizers are used for diversity reception to improve the quality of wireless communication systems [21]. [22, 23] proposed multichannel DFE receiver for underwater acoustic communications. Received signals are processed by a bank of adaptive linear filters that jointly perform matched filtering and feed-forward equalization. Adaptive phase synchronization is then performed on each branch before the signals are combined and passed to a single DFE feedback and decision loop. In [24], the received signal is split into two streams. One stream is equalized after time reversal (TR) combining. Another stream is flipped in the time domain, processed by TR-combining and equalization, and then flipped in the time domain again. After processing, each stream is conducted diversity combining.”

On page 20, add: S1 Appendix.

Reviewer#2, Concern # 8: There are too many spelling and grammar mistakes in the paper. It needs proper spelling and grammar checking.

Author response: Thank you for your valuable suggestion. We apologize for so many spelling and grammar mistakes. We have thoroughly reviewed the entire paper and made corrections to the spelling and grammar mistakes.

Reviewer#2, Concern # 9: Most mentioned Figures includes small details; so they are not clear for the readers to distinguish the difference and the performance improvement of image enhancement methods. Add full description for all the results.

Author response: Thank you for your valuable suggestion. We have added comparison experiments with the algorithm proposed in ref [22, 23] in both the simulation and lake trial sections. The simulation results are shown in Fig 13. And to provide a better description of the data in the figure, we have extracted the key data and presented them in the form of tables. At the same time, we have provided a detailed description of the results comparing our proposed algorithm with the results of the comparison algorithms.

On page 14 line 408-421:

“The simulation results are shown in Fig 13. The number of receiving antennas also represents the number of diversity.

In order to provide a clearer description of the data in the figures, we have compiled the data into Table 3 and Table 4.

From Fig 13, Table 3 and Table 4, it can be observed that the system’s BER decreases as the number of receiving antennas increases. Compared to the point-to-point communication link system, DFE-PLL-MSIRC can achieve a diversity gain of more than 5.2 dB, while the comparison algorithm can achieve a diversity gain of more than 2.3 dB. The proposed DFE-PLL-MSIRC can obtain about 3 dB more diversity gain than the comparison algorithm. When SNR is 0 dB, the BER of DFE-PLL-MSIRC can be reduced by at least one order of magnitude, while the comparison algorithm can be reduced by at least 0.4 order of magnitude. Overall, the BER of DFE-PLL-MSIRC is lower than that of the comparison algorithm.”

On page 15 line 439-453:

“In order to provide a clearer description of the data in the figures, we have compiled the data into Table 5. 

As seen from Fig 14, with the gradual deterioration of the marine environment, the BER of the same diversity also gradually increases. The minimum diversity order required for the system to maintain communication is shown in Table 5. We can observe that in the cases of marine environment 1 and 3, the comparison algorithm requires a higher diversity order than DFE-PLL-MSIRC, thereby validating that DFE-PLL-MSIRC is more efficient than the comparison algorithm.”

On page 17 line 498-510:

“It can be seen from Fig 18 and Fig 19 that the proposed system’s constellation diagrams are more convergent with the increase of the diversity number. The constellation map of DFE-PLL-MSIRC tends to approach the constellation points {1, 1}, {1, −1}, {−1, 1}, and {−1, −1} of the original QPSK signal, more so than the constellation map of the comparison algorithm. It can be seen from Table 7 that in the case of the transceiver spacing of 500m, compared with point-to-point communication, the BER of two-point diversity combination can be reduced by an order of magnitude, and the BER of three-point diversity combination can be reduced to 0. In the case of the transceiver spacing of 1000 m, compared with point-to-point communication, the BER of two-point diversity combination can be reduced by 0.5 orders of magnitude, and the BER of three-point diversity combination can be reduced by 1 order of magnitude. Regardless of whether the transmission distance is 500m or 1000m, the BER of DFE-PLL-MSIRC is lower than that of the comparison algorithm.”

Reviewer#2, Concern # 10: In conclusion, you can show briefly the percentage of improvement of image enhancement methods in terms of quantitative evaluation measures.

Author response: Thank you for your valuable suggestion. We have revised the conclusion section and provided a brief presentation of the improvement percentages of the proposed spatial diversity equalizer and the comparison algorithms.

On page 18 line 519-524, add: “Simulation and lake experiments results show that, compared to the point-to-point communication link system, DFE-PLL-MSIRC can achieve a diversity gain of more than 5.2 dB and obtain about 3 dB diversity gain more than the comparison algorithm. Moreover, the BER of DFE-PLL-MSIRC can be reduced by at least one order of magnitude, which is lower by at least 0.6 order of magnitude compared to the comparison algorithm.”

Reviewer#2, Concern # 11: What are the trends and future direction of applying those proposed method? Please give your suggestions.

Author response: Thank you for your valuable suggestion. We have added the discusses about the trends and future directions of applying the proposed method at the end of the conclusion.

On page 19 line 531-536, add: “Further research on the distributed processing mechanism of underwater acoustic signals for spatial information networks, effectively extending the coverage of spatial information networks to the maritime domain, and achieving global stereo coverage of the space-sky-ground-sea integrated space information network is our future trend and direction.”

Reviewer#2, Concern # 12: Add some recently papers (2020-2023) in the manuscript.

Author response: Thank you for your valuable suggestion. We updated the manuscript by adding new references between the years 2020 to date.

On page 1 line 3-8: add [3-6] in “By utilizing the features of broad coverage, fade resistance, and minimal required switching provided by the distributed antenna systems (DAS) [1] [2], underwater sensor nodes collect and transmit data through point-to-point and multipoint-to-multipoint communication to construct a distributed underwater sensor network for the effective transmission of information [3-6].”

On page 4 line 159-162: add [41] in “In most scenarios, the underwater acoustic channel can be considered a slowly varying multipath channel with coherence over time. The sound rays emitted from the source travel to the receiver through multiple routes, and the received sound field combines all the arriving rays [41].”

[3] Sathish K, Ravikumar C, Srinivasulu A, Rajesh A, Oyerinde OO, et al. Performance and Improvement Analysis of the Underwater WSN Using a Diverse Routing Protocol Approach. Journal of Computer Networks and Communications. 2022;2022.

[4] Sathish K, Hamdi M, Chinthaginjala R, Pau G, Ksibi A, Anbazhagan R, et al. Reliable Data Transmission in Underwater Wireless Sensor Networks Using a Cluster-Based Routing Protocol Endorsed by Member Nodes. Electronics. 2023;12(6):1287.

[5] Sathish K, CV R, Ab Wahab MN, Anbazhagan R, Pau G, Akbar MF. Underwater Wireless Sensor Networks Performance Comparison Utilizing Telnet and Superframe. Sensors. 2023;23(10):4844.

[6] Sathish K, Ravikumar CV, Rajesh A, Pau G. Underwater wireless sensor network performance analysis using diverse routing protocols. Journal of Sensor and Actuator Networks. 2022;11(4):64.

[41] Zhou M, Zhang H, Lv T, Li H, Xiang D, Huang S, et al. Underwater acoustic channel modeling under different shallow seabed topography and sediment environment. In: OCEANS 2022-Chennai. IEEE; 2022. p. 1–7.

Thanks again for your advice and I hope to learn more from you.

This part is the response to the first reviewer's comments in the first round of revisions. In order to maintain the completeness of the response letter, it has been retained at the end.

Reviewer#1, Concern # 1: The way of redacting the sentence: "... systems [9]. [10]conducted a ..." in page 2/15, first paragraph is not very fortunate. It is suggested to improve the redaction and insert a space to separate words while typing. For example as can be seen in: "... [17] design rate ..." in the second paragraph on page 2/15.

Author response: We are sorry for this editing error. We have improved the redaction and inserted a space to separate words while typing. 

On page 2/17, first paragraph: "... systems [9]. [10] conducted a ...".

Reviewer#1, Concern # 2: The way of redacting the sentence: "... channels.In practice, ..." in page 2/15, first paragraph is not very fortunate. It is suggested to improve the redaction and insert a space to separate words after a period.

Author response: We are sorry for this editing error. We have improved the redaction and inserted a space to separate words after a period. 

On page 2/17, first paragraph: "... channels. In practice, ...".

Reviewer#1, Concern # 3: It is suggested to improve the same in: "... [13]proposed a ..."; "... [14]studies a ..."; "... [16]cast space–time codes ..."; "... [19]proposed ..."; "... advantages:The phase ..."; "... compensation.The combining ...".

Author response: We are sorry for this editing error. We have improved the redaction and inserted a space to separate words while typing. Since we have added new references, the numbers of references have also been updated.

On page 2/17, first paragraph: "To address this problem, [13] proposed a ..."; "... [14] studies a ...".

On page 2/17, second paragraph: "... [18] cast space–time codes ..."; "... The proposed transmission scheme [20] combines the ..."; "... [21] proposed ...".

On page 3/17: "... advantages: The phase ..."; "... compensation. The combining ...".

Reviewer#1, Concern # 4: In page 3/15, section "BELLHOP Gaussian beam tracking model", first paragraph, the redaction is not very fortunate: "... sound field research22 ...". It is suggested to improve the redaction.

Author response: We are sorry for this editing error. We have improved the redaction of Reference 22. Since we have added new references, the numbers of references have also been updated.

On page 4/17, first paragraph: "... sound field research [27] ...".

Reviewer#1, Concern # 5: It is suggested to improve the redaction in: "... model [24]is shown..." in page 3/15, section "BELLHOP Gaussian beam tracking model", second paragraph.

Author response: We are sorry for this editing error. We have improved the redaction and inserted a space to separate words while typing. Since we have added new references, the numbers of references have also been updated.

On page 4/17, second paragraph: "... model [29] is shown ...".

Reviewer#1, Concern # 6: In page 4/15, it is not clear if conceptually the authors are referring to h(t) as the impulse response function, defined in the equation (1), but at the same time, as h(t) to the transfer function. This could be improved.

Author response: Thank you for your valuable suggestions. I'm sorry for the conceptual confusion. If the underwater acoustic channel is regarded as a system, the input signal passes through the underwater acoustic channel to obtain the output signal. In the time domain, the impulse response is used to describe the characteristics of the system. That is, the input signal is convoluted with the impulse response to obtain the output signal of the system. In the frequency domain, the transfer function is used to describe the characteristics of the system. That is, the input signal is multiplied by the transfer function of the system to obtain the output signal of the system. In this paper, we model the underwater acoustic channel in the time domain. Therefore, h(t) should be described more accurately by impulse response. 

We have corrected the transfer function h(t) to the impulse response h(t) on page 4/17, section " Underwater acoustic channel impulse response h(t)".

Reviewer#1, Concern # 7: It is suggested to improve the redaction in page 4/15: "... based on a cross-media buoys network. Based on the ...". In the same line, "based on" is used two times.

Author response: Thank you for your valuable suggestions. We have rewritten this part for better readability.

On page 5/17, first paragraph: "This paper constructs a DUA-DCS based on a cross-media distributed buoys network. Because the multiple underwater acoustic communication links distributed in a certain area have the characteristics of mutual statistical independence, the system uses the buoys network with motorized deployment to convert the ’point-to-point’ variable parameter channel into a ’point-to-multipoint’ or ’multi-point-to-point’ quasi-constant parameter stationary channel.".

Reviewer#1, Concern # 8: In equation (4), page 6/15, the parameter "e" is not defined. It is suggested to improve the redaction.

Author response: Thank you for your valuable suggestions. We have defined the parameter "e" and updated the manuscript.

On page 7/17, fourth paragraph: "e(i) denotes the error between the desired output value d(i) of the equalizer and the decision value y ^(i) at the time i."

Reviewer#1, Concern # 9: It is suggested to improve the redaction in: "... Initialization:ω (0) = ..."; "... Among them,y (n) = ..." in page 7/15, and insert spaces to separate words while typing.

Author response: We are sorry for this editing error. We have improved the redaction and inserted a space to separate words while typing.

On page 7/17: "... Initialization: ω (0) = ..."; "... Among them, y (n) = ...".

Reviewer#1, Concern # 10: In page 7/15, "... y_b (n)=[y(n-1),⋯,y(n-N_3 )]. ...", the parameter "N_3" is not defined, or the question is if it corresponds to "N_2", the order of FBF. This could be improved.

Author response: Thank you for your valuable suggestions. The parameter "N_3" corresponds to "N_2", that is, the order of FBF. We have corrected it in the manuscript.

On page 7/17: "y_b (n)=[y(n-1),⋯,y(n-N_2 )]".

Reviewer#1, Concern # 11: In page 8/15, the style of the equation (16) is not very lucky: "... h_(j,i) (t)=h_(j,i) (t+T)=h_(j,i),t=1,2,⋯p ". This one could be improved.

Author response: Thank you for your valuable suggestions. We have modified equation (16) according to the correct format. 

On page 10/17, equation (16): "h_(j,i) (t)=h_(j,i) (t+T)=h_(j,i) 

Where t=1,2,⋯p."

Reviewer#1, Concern # 12: In page 8/15, the style of the equation (18) is not very lucky: "... xi = argmind2...". This one could be improved.

Author response: Thank you for your valuable suggestions. We have modified equation (18) according to the correct format. 

On page 10/17, equation (18): " x ^_i=arg min┬(x ^_i ϵS)⁡〖d^2 (x ~_i,x ^_i)〗+(∑_(i=1)^(N_i)▒∑_(j=1)^(N_t)▒〖|h_(j,i) |^2-1)|x_i |^2 〗

Where i=1,2,⋯k."

Reviewer#1, Concern # 13: In page 11/15, it is suggested to improve the redaction both in: "... Case 2:Verify ... ", and "... Case 3:This ..." Insert spaces to separate words while typing.

Author response: We are sorry for this editing error. We have improved the redaction and inserted a space to separate words while typing. 

On page 12/17, second paragraph: "... Case 2: Verify ... ".

On page 12/17, fourth paragraph: "... Case 3: This ...".

Reviewer#1, Concern # 14: For consistency in writing, to refer to the equations, the authors use on page 8/15: "... It satisfies Eq (12)...". However, on page 11/15, the authors employ: "...as Eq 14 and Eq 15. ...". It is suggested to improve the redaction to be consistent in writing.

Author response: Thank you for your valuable suggestions. For consistency in writing, we have corrected "...as Eq 14 and Eq 15 ..." to be consistent with "... It satisfies Eq (12) ...".

On page 12/17, fourth paragraph: " Case 3: This simulation is to verify the influence of the transmit antennas’ number on the system performance in the downlink under the same environmental parameters. The complex orthogonal transmission matrixes of two transmitting antennas and three transmitting antennas are designed as Eq (14) and Eq (15)."

Reviewer#1, Concern # 15: In page 12/15, section Lake experiments and results. It would be opportune to know if the authors before the field measurements have carried out, or not, a calibration procedure of the hydrophones and of all the experimental setup. If some calibration procedure was carried out, it would be appropriated that the authors describe it.

Author response: Thank you for your valuable suggestions. Before the field measurements, we carried out a calibration procedure of the hydrophones and all the experimental setups. We updated the manuscript by describing this part in detail.

On page 12/17: Add "Before the lake experiments, the hydrophones, and other experimental equipment have been professionally calibrated by the manufacturer. In the meantime, we have completed tasks such as testing electrical connections and ensuring communication link availability in a water tank environment. The transmitter sends a QPSK modulation signal through the transducer connected to the underwater acoustic communication apparatus module. The receiver collects underwater acoustic signals through a transducer connected to a digital collector. The obtained underwater acoustic signal is processed by Matlab. If the data can be successfully demodulated, it is proved that the communication link is passable and the electrical connection is correct."

Reviewer#1, Concern # 16: In page 12/15, it is suggested to improve the redaction in: "... Fig 15.The equipment ..." and insert space to separate words while typing.

Author response: We are sorry for this editing error. We have improved the redaction and inserted a space to separate words while typing. 

On page 13/17, first paragraph: "... Fig. 15. The equipment ...".

Reviewer#1, Concern # 17: Table 3, in page 12/15: The way of presenting the results in Table 3 is not very fortunate. It is suggested to improve the presentation.

Author response: Thank you for your valuable suggestions. We modified Table 3 according to the standard of the three-line table. To make readers better understand, Table 3 is explained in detail in the manuscript. At the same time, to maintain the consistency of the format of all tables, we also modified Table 1, Table 2, and Table 4 to the format of the three-line table.

On page 13/17, first paragraph: add "On the signal transmitting ship, the personal computer (PC) is connected to the underwater acoustic communication apparatus, and then the underwater acoustic communication apparatus is connected to the transducer. On the signal-receiving ship, the PC is connected to the digital collector, and then the digital collector is connected to the transducer. Among them, YOKOGAWA SL1000, HIOKI MR6000, and Altai represent the models of digital collectors."

Reviewer#1, Concern # 18: Throughout the manuscript, each of the "Fig" abbreviations should carry a period. This is, for example, "Fig. 15". This one could be improved.

Author response: Thank you very much for your valuable comments, we have modified each of the "Fig" to "Fig." according to your comments.

Reviewer#1, Concern # 19: In the references, the authors only show one article published in 2019. No article is reviewed between the years 2020 to date. This could be improved. It is suggested that the authors update the references according to the state of the art.

Author response: Thank you for your valuable suggestion. We updated the manuscript by adding new references between the years 2020 to date.

On page 2/17, first paragraph: add [15] in “In [15], the received signal is split into two streams. One stream is equalized after time reversal (TR) combining. Another stream is flipped in the time domain, processed by TR-combining and equalization, and then flipped in the time domain again. After processing, each stream is conducted diversity combining.”

On page 2/17, first paragraph: add [16] in “[16] employed spatial diversity and antenna beamforming methods to significantly improve the signal quality performance of wireless networks and enhance the reliability by decreasing the bit error probability.”

On page 3/17, first paragraph: add [24] in “In [24], joint time-reversal space-time block coding and adaptive equalization filtered multitone underwater acoustic communication method was proposed. The effectiveness of the proposed method is verified by simulation analysis and real experimental data collected from an indoor pool communication trial.”

On page 3/17, first paragraph: add [25] in “[25] proposed a generalized space-time block coded spatial modulation scheme for open-loop massive multiple-input and multiple-output downlink communication systems. The information bits are divided into multiple groups with each group modulated by the spatial modulation, where the SM symbols are invoked for OSTBC and quasi-orthogonal STBC structures.”

On page 3/17, first paragraph: add [26] in “[26] proposed space-time block coded non-orthogonal multiple access (STBC-NOMA) for underwater acoustic sensor networks (UASNs) to improve reliability by exploiting transmit diversity and spectral efficiency. Results show that STBC-NOMA can significantly enhance the performance of UASNs without the need for prior channel state information status at the transmitter.”

[15] Kim H, Choi KH, Choi JW, Bae HS. Bidirectional Equalization for Long-Range Underwater Acoustic Communication in BLAC18. In: 2019 Eleventh International Conference on Ubiquitous and Future Networks (ICUFN). IEEE; 2019. p. 52–53.

[16] Mehta R. Optimal receive beamforming in spatial antenna diversity system using evolutionary genetic algorithm. Array. 2021; 10:100053.

[24] Sun L, Yan M, Li H, Xu Y. Joint Time-Reversal Space-Time Block Coding and Adaptive Equalization for Filtered Multitone Underwater Acoustic Communications. Sensors. 2020;20(2):379.

[25] Xiao L, Chen D, Hemadeh I, Xiao P, Jiang T. Generalized space-time block coded spatial modulation for open-loop massive MIMO downlink communication systems. IEEE Transactions on Communications. 2020;68(11):6858–6871.

[26] Goutham V, Harigovindan V. Space–time block coded non-orthogonal multiple access for performance enhancement of underwater acoustic sensor networks. ICT Express. 2022;8(1):117–123.

Thanks again for your advice and I hope to learn more from you.

---

## [Decision Letter · Decision Letter 2]

16 Nov 2023

PONE-D-22-31101R2Spatial diversity processing mechanism based on the distributed underwater acoustic communication systemPLOS ONE

Dear Dr. Zhou,

Thank you for submitting your manuscript to PLOS ONE. After careful consideration, we feel that it has merit but does not fully meet PLOS ONE’s publication criteria as it currently stands. Therefore, we invite you to submit a revised version of the manuscript that addresses the points raised during the review process.

We look forward to receiving your revised manuscript.

Kind regards,

Hanna Landenmark

Staff Editor

PLOS ONE

on behalf of 

Ravikumar Chinthaginjala

Academic Editor

PLOS ONE

Journal Requirements:

**Additional Editor Comments:**

We noted the following outstanding items:

1) Please thoroughly copyedit the manuscript before it can be considered for publication. You may use a professional editing service, or a colleague.

2) Please update your Data availability statement to indicate where the data used can be accessed by other researchers. 

Reviewers' comments:

Reviewer's Responses to Questions

**Comments to the Author**

1. If the authors have adequately addressed your comments raised in a previous round of review and you feel that this manuscript is now acceptable for publication, you may indicate that here to bypass the “Comments to the Author” section, enter your conflict of interest statement in the “Confidential to Editor” section, and submit your "Accept" recommendation.

Reviewer #2: All comments have been addressed

Reviewer #3: All comments have been addressed

2. Is the manuscript technically sound, and do the data support the conclusions?

Reviewer #2: Yes

Reviewer #3: Yes

3. Has the statistical analysis been performed appropriately and rigorously? 

Reviewer #2: Yes

Reviewer #3: Yes

4. Have the authors made all data underlying the findings in their manuscript fully available?

Reviewer #2: Yes

Reviewer #3: Yes

5. Is the manuscript presented in an intelligible fashion and written in standard English?

Reviewer #2: Yes

Reviewer #3: Yes

6. Review Comments to the Author

Reviewer #2: Actually, after the revision, the current revised manuscript is somehow better than the previous one. Check with journal template.

All the best for the authors

Reviewer #3: Dear Author

all the reviewer comments have been incorporated.

pleased to inform that the paper can be accepted.

7. PLOS authors have the option to publish the peer review history of their article (what does this mean?). If published, this will include your full peer review and any attached files.

Reviewer #2: No

Reviewer #3: No

---

## [Author Response · Author response to Decision Letter 2]

29 Nov 2023

Original Manuscript ID: PONE-D-22-31101R2 

Original Article Title: “Spatial diversity processing mechanism based on the distributed underwater acoustic communication system”

To: PLOS ONE Editor

Re: Response to reviewers

Dear Editor,

Thank you for allowing a resubmission of our manuscript, with an opportunity to address the reviewers’ comments.

We are uploading (a) our point-by-point response to the comments (below) (Response to Reviewers), (b) an updated manuscript with yellow highlighting indicating changes (Revised Manuscript with Track Changes), and (c) a clean updated manuscript without highlights (Manuscript).

Best regards,

<Manli Zhou> et al.

Concern # 1: Response: Thank you for your valuable suggestion. We have updated the financial disclosure and revised the cover letter. 

Concern # 2: Journal Requirements: Please review your reference list to ensure that it is complete and correct. If you have cited papers that have been retracted, please include the rationale for doing so in the manuscript text, or remove these references and replace them with relevant current references. Any changes to the reference list should be mentioned in the rebuttal letter that accompanies your revised manuscript. If you need to cite a retracted article, indicate the article’s retracted status in the References list and also include a citation and full reference for the retraction notice.

Author response: We have reviewed the list of my references. We did not cite any retracted papers. And there have been no changes to the reference list.

Concern # 3: Additional Editor Comments: We noted the following outstanding items: 1) Please thoroughly copyedit the manuscript before it can be considered for publication. You may use a professional editing service, or a colleague.

Author response: We have proofread the manuscript thoroughly. We have thoroughly proofread the manuscript, diligently correcting grammar and formatting errors to enhance its readability. 

Concern # 4: 2) Please update your Data availability statement to indicate where the data used can be accessed by other researchers.

Author response: We have updated my Data availability statement. All data files are available from the figshare database. (https://doi.org/10.6084/m9.figshare.23995809.v2).

 

This part is the response to the reviewer 's comments. In order to maintain the completeness of the response letter, it has been retained at the end.

Reviewer#1, Concern # 1: The way of redacting the sentence: "... systems [9]. [10]conducted a ..." in page 2/15, first paragraph is not very fortunate. It is suggested to improve the redaction and insert a space to separate words while typing. For example as can be seen in: "... [17] design rate ..." in the second paragraph on page 2/15.

Author response: We are sorry for this editing error. We have improved the redaction and inserted a space to separate words while typing. 

On page 2/17, first paragraph: "... systems [9]. [10] conducted a ...".

Reviewer#1, Concern # 2: The way of redacting the sentence: "... channels.In practice, ..." in page 2/15, first paragraph is not very fortunate. It is suggested to improve the redaction and insert a space to separate words after a period.

Author response: We are sorry for this editing error. We have improved the redaction and inserted a space to separate words after a period. 

On page 2/17, first paragraph: "... channels. In practice, ...".

Reviewer#1, Concern # 3: It is suggested to improve the same in: "... [13]proposed a ..."; "... [14]studies a ..."; "... [16]cast space–time codes ..."; "... [19]proposed ..."; "... advantages:The phase ..."; "... compensation.The combining ...".

Author response: We are sorry for this editing error. We have improved the redaction and inserted a space to separate words while typing. Since we have added new references, the numbers of references have also been updated.

On page 2/17, first paragraph: "To address this problem, [13] proposed a ..."; "... [14] studies a ...".

On page 2/17, second paragraph: "... [18] cast space–time codes ..."; "... The proposed transmission scheme [20] combines the ..."; "... [21] proposed ...".

On page 3/17: "... advantages: The phase ..."; "... compensation. The combining ...".

Reviewer#1, Concern # 4: In page 3/15, section "BELLHOP Gaussian beam tracking model", first paragraph, the redaction is not very fortunate: "... sound field research22 ...". It is suggested to improve the redaction.

Author response: We are sorry for this editing error. We have improved the redaction of Reference 22. Since we have added new references, the numbers of references have also been updated.

On page 4/17, first paragraph: "... sound field research [27] ...".

Reviewer#1, Concern # 5: It is suggested to improve the redaction in: "... model [24]is shown..." in page 3/15, section "BELLHOP Gaussian beam tracking model", second paragraph.

Author response: We are sorry for this editing error. We have improved the redaction and inserted a space to separate words while typing. Since we have added new references, the numbers of references have also been updated.

On page 4/17, second paragraph: "... model [29] is shown ...".

Reviewer#1, Concern # 6: In page 4/15, it is not clear if conceptually the authors are referring to h(t) as the impulse response function, defined in the equation (1), but at the same time, as h(t) to the transfer function. This could be improved.

Author response: Thank you for your valuable suggestions. I'm sorry for the conceptual confusion. If the underwater acoustic channel is regarded as a system, the input signal passes through the underwater acoustic channel to obtain the output signal. In the time domain, the impulse response is used to describe the characteristics of the system. That is, the input signal is convoluted with the impulse response to obtain the output signal of the system. In the frequency domain, the transfer function is used to describe the characteristics of the system. That is, the input signal is multiplied by the transfer function of the system to obtain the output signal of the system. In this paper, we model the underwater acoustic channel in the time domain. Therefore, h(t) should be described more accurately by impulse response. 

We have corrected the transfer function h(t) to the impulse response h(t) on page 4/17, section " Underwater acoustic channel impulse response h(t)".

Reviewer#1, Concern # 7: It is suggested to improve the redaction in page 4/15: "... based on a cross-media buoys network. Based on the ...". In the same line, "based on" is used two times.

Author response: Thank you for your valuable suggestions. We have rewritten this part for better readability.

On page 5/17, first paragraph: "This paper constructs a DUA-DCS based on a cross-media distributed buoys network. Because the multiple underwater acoustic communication links distributed in a certain area have the characteristics of mutual statistical independence, the system uses the buoys network with motorized deployment to convert the ’point-to-point’ variable parameter channel into a ’point-to-multipoint’ or ’multi-point-to-point’ quasi-constant parameter stationary channel.".

Reviewer#1, Concern # 8: In equation (4), page 6/15, the parameter "e" is not defined. It is suggested to improve the redaction.

Author response: Thank you for your valuable suggestions. We have defined the parameter "e" and updated the manuscript.

On page 7/17, fourth paragraph: "e(i) denotes the error between the desired output value d(i) of the equalizer and the decision value y ^(i) at the time i."

Reviewer#1, Concern # 9: It is suggested to improve the redaction in: "... Initialization:ω (0) = ..."; "... Among them,y (n) = ..." in page 7/15, and insert spaces to separate words while typing.

Author response: We are sorry for this editing error. We have improved the redaction and inserted a space to separate words while typing.

On page 7/17: "... Initialization: ω (0) = ..."; "... Among them, y (n) = ...".

Reviewer#1, Concern # 10: In page 7/15, "... y_b (n)=[y(n-1),⋯,y(n-N_3 )]. ...", the parameter "N_3" is not defined, or the question is if it corresponds to "N_2", the order of FBF. This could be improved.

Author response: Thank you for your valuable suggestions. The parameter "N_3" corresponds to "N_2", that is, the order of FBF. We have corrected it in the manuscript.

On page 7/17: "y_b (n)=[y(n-1),⋯,y(n-N_2 )]".

Reviewer#1, Concern # 11: In page 8/15, the style of the equation (16) is not very lucky: "... h_(j,i) (t)=h_(j,i) (t+T)=h_(j,i),t=1,2,⋯p ". This one could be improved.

Author response: Thank you for your valuable suggestions. We have modified equation (16) according to the correct format. 

On page 10/17, equation (16): "h_(j,i) (t)=h_(j,i) (t+T)=h_(j,i) 

Where t=1,2,⋯p."

Reviewer#1, Concern # 12: In page 8/15, the style of the equation (18) is not very lucky: "... xi = argmind2...". This one could be improved.

Author response: Thank you for your valuable suggestions. We have modified equation (18) according to the correct format. 

On page 10/17, equation (18): " x ^_i=arg min┬(x ^_i ϵS)⁡〖d^2 (x ~_i,x ^_i)〗+(∑_(i=1)^(N_i)▒∑_(j=1)^(N_t)▒〖|h_(j,i) |^2-1)|x_i |^2 〗

Where i=1,2,⋯k."

Reviewer#1, Concern # 13: In page 11/15, it is suggested to improve the redaction both in: "... Case 2:Verify ... ", and "... Case 3:This ..." Insert spaces to separate words while typing.

Author response: We are sorry for this editing error. We have improved the redaction and inserted a space to separate words while typing. 

On page 12/17, second paragraph: "... Case 2: Verify ... ".

On page 12/17, fourth paragraph: "... Case 3: This ...".

Reviewer#1, Concern # 14: For consistency in writing, to refer to the equations, the authors use on page 8/15: "... It satisfies Eq (12)...". However, on page 11/15, the authors employ: "...as Eq 14 and Eq 15. ...". It is suggested to improve the redaction to be consistent in writing.

Author response: Thank you for your valuable suggestions. For consistency in writing, we have corrected "...as Eq 14 and Eq 15 ..." to be consistent with "... It satisfies Eq (12) ...".

On page 12/17, fourth paragraph: " Case 3: This simulation is to verify the influence of the transmit antennas’ number on the system performance in the downlink under the same environmental parameters. The complex orthogonal transmission matrixes of two transmitting antennas and three transmitting antennas are designed as Eq (14) and Eq (15)."

Reviewer#1, Concern # 15: In page 12/15, section Lake experiments and results. It would be opportune to know if the authors before the field measurements have carried out, or not, a calibration procedure of the hydrophones and of all the experimental setup. If some calibration procedure was carried out, it would be appropriated that the authors describe it.

Author response: Thank you for your valuable suggestions. Before the field measurements, we carried out a calibration procedure of the hydrophones and all the experimental setups. We updated the manuscript by describing this part in detail.

On page 12/17: Add "Before the lake experiments, the hydrophones, and other experimental equipment have been professionally calibrated by the manufacturer. In the meantime, we have completed tasks such as testing electrical connections and ensuring communication link availability in a water tank environment. The transmitter sends a QPSK modulation signal through the transducer connected to the underwater acoustic communication apparatus module. The receiver collects underwater acoustic signals through a transducer connected to a digital collector. The obtained underwater acoustic signal is processed by Matlab. If the data can be successfully demodulated, it is proved that the communication link is passable and the electrical connection is correct."

Reviewer#1, Concern # 16: In page 12/15, it is suggested to improve the redaction in: "... Fig 15.The equipment ..." and insert space to separate words while typing.

Author response: We are sorry for this editing error. We have improved the redaction and inserted a space to separate words while typing. 

On page 13/17, first paragraph: "... Fig. 15. The equipment ...".

Reviewer#1, Concern # 17: Table 3, in page 12/15: The way of presenting the results in Table 3 is not very fortunate. It is suggested to improve the presentation.

Author response: Thank you for your valuable suggestions. We modified Table 3 according to the standard of the three-line table. To make readers better understand, Table 3 is explained in detail in the manuscript. At the same time, to maintain the consistency of the format of all tables, we also modified Table 1, Table 2, and Table 4 to the format of the three-line table.

On page 13/17, first paragraph: add "On the signal transmitting ship, the personal computer (PC) is connected to the underwater acoustic communication apparatus, and then the underwater acoustic communication apparatus is connected to the transducer. On the signal-receiving ship, the PC is connected to the digital collector, and then the digital collector is connected to the transducer. Among them, YOKOGAWA SL1000, HIOKI MR6000, and Altai represent the models of digital collectors."

Reviewer#1, Concern # 18: Throughout the manuscript, each of the "Fig" abbreviations should carry a period. This is, for example, "Fig. 15". This one could be improved.

Author response: Thank you very much for your valuable comments, we have modified each of the "Fig" to "Fig." according to your comments.

Reviewer#1, Concern # 19: In the references, the authors only show one article published in 2019. No article is reviewed between the years 2020 to date. This could be improved. It is suggested that the authors update the references according to the state of the art.

Author response: Thank you for your valuable suggestion. We updated the manuscript by adding new references between the years 2020 to date.

On page 2/17, first paragraph: add [15] in “In [15], the received signal is split into two streams. One stream is equalized after time reversal (TR) combining. Another stream is flipped in the time domain, processed by TR-combining and equalization, and then flipped in the time domain again. After processing, each stream is conducted diversity combining.”

On page 2/17, first paragraph: add [16] in “[16] employed spatial diversity and antenna beamforming methods to significantly improve the signal quality performance of wireless networks and enhance the reliability by decreasing the bit error probability.”

On page 3/17, first paragraph: add [24] in “In [24], joint time-reversal space-time block coding and adaptive equalization filtered multitone underwater acoustic communication method was proposed. The effectiveness of the proposed method is verified by simulation analysis and real experimental data collected from an indoor pool communication trial.”

On page 3/17, first paragraph: add [25] in “[25] proposed a generalized space-time block coded spatial modulation scheme for open-loop massive multiple-input and multiple-output downlink communication systems. The information bits are divided into multiple groups with each group modulated by the spatial modulation, where the SM symbols are invoked for OSTBC and quasi-orthogonal STBC structures.”

On page 3/17, first paragraph: add [26] in “[26] proposed space-time block coded non-orthogonal multiple access (STBC-NOMA) for underwater acoustic sensor networks (UASNs) to improve reliability by exploiting transmit diversity and spectral efficiency. Results show that STBC-NOMA can significantly enhance the performance of UASNs without the need for prior channel state information status at the transmitter.”

[15] Kim H, Choi KH, Choi JW, Bae HS. Bidirectional Equalization for Long-Range Underwater Acoustic Communication in BLAC18. In: 2019 Eleventh International Conference on Ubiquitous and Future Networks (ICUFN). IEEE; 2019. p. 52–53.

[16] Mehta R. Optimal receive beamforming in spatial antenna diversity system using evolutionary genetic algorithm. Array. 2021; 10:100053.

[24] Sun L, Yan M, Li H, Xu Y. Joint Time-Reversal Space-Time Block Coding and Adaptive Equalization for Filtered Multitone Underwater Acoustic Communications. Sensors. 2020;20(2):379.

[25] Xiao L, Chen D, Hemadeh I, Xiao P, Jiang T. Generalized space-time block coded spatial modulation for open-loop massive MIMO downlink communication systems. IEEE Transactions on Communications. 2020;68(11):6858–6871.

[26] Goutham V, Harigovindan V. Space–time block coded non-orthogonal multiple access for performance enhancement of underwater acoustic sensor networks. ICT Express. 2022;8(1):117–123.

 

Reviewer#2, Concern # 1: It is better to revise or suggest another appropriate title for this Manuscript to reflect the details of the work.

Author response: Thank you for your valuable suggestion. We have revised the title of the manuscript to “Spatial diversity processing mechanism based on the distributed underwater acoustic communication system”.

Reviewer#2, Concern # 2: The abstract does not communicate effectively; it should be changed.

Author response: Thank you for your valuable suggestion. We have revised the abstract to make it clearer and more readable.

On page 1, first paragraph: “To address the problem of unreliable single-link underwater acoustic communication caused by large signal delays and strong multipath effects in shallow water environments, this paper proposes a distributed underwater acoustic diversity communication system (DUA-DCS). DUA-DCS employs a maneuverable distributed cross-medium buoy network to form multiple distributed, non-coherent, and parallel communication links. In the uplink, a receiving diversity processing mechanism of joint decision feedback equalizer embedded phase-locked loop and maximum signal-to-interference ratio combining (DFE-PLL-MSIRC) is proposed to achieve waveform-level diversity combining of underwater signals. A phase-locked loop module is embedded in each branch of the decision feedback equalizer to eliminate the residual frequency and phase errors after Doppler compensation. Meanwhile, the combining coefficients are determined based on the maximum signal-to-interference ratio criterion, taking into account the residual inter-symbol interference after equalization, resulting in efficient and accurate computation. Additionally, the combined decision values are fed back to the feedback filters in each branch to ensure more accurate feedback output. Simulation and lake experiment results demonstrate that, compared to the single-link communication system, DFE-PLL-MSIRC can achieve a diversity gain of more than 5.2 dB and obtain about 3 dB more diversity gain than the comparison algorithm. And the BER of DFE-PLL-MSIRC can be reduced by at least one order of magnitude, which is lower by at least 0.6 order of magnitude compared to the comparison algorithm. In the downlink, a transmitting diversity processing mechanism of complex orthogonal space-time block coding (COSTBC) is proposed. By utilizing a newly designed generalized complex orthogonal transmission matrix, complete transmission diversity can be achieved at the coding rate of 3/4. Compared to the single-link communication system, the system can achieve a diversity gain of more than 6 dB.” ________________________________________

Reviewer#2, Concern # 3: The paper organization should be revised with more details about the proposed work in this manuscript.

Author response: Thank you for your valuable suggestion. In order to provide a more detailed description of the work proposed in the manuscript, we have revised the organization structure of the manuscript and included the following content:

On page 4 line 134-138, add: “The remainder of this manuscript is organized as follows. Section 2 introduces the underwater acoustic channel model. Section 3 introduces the system architecture. Section 4 introduces the spatial diversity processing mechanism. The results of the simulations and lake experiments are presented in Section 5 and 6, respectively. In Section 7, we finally conclude.”

On page 6 line 225-252, we have added the subsubsection “Decision feedback equalizer”.

On page 8 line 269-290, we have added the subsubsection “Phase-locked loop”.

On page 9 line 291-336, we have revised the subsubsection “DFE-PLL-MSIRC”.

Reviewer#2, Concern # 4: In particular, the paper’s contribution is unclear compared to existing works in the literature, and there lacks a necessary comparison with the existing image enhancement methods. Thus, the current version cannot be recommended for acceptance.

Author response: Thank you for your valuable suggestion. I 'm sorry to make you confused. The main theme of this paper is not about image enhancement. This paper's primary focus is constructing a distributed underwater acoustic diversity communication system to address the issue of unreliability in single-link underwater acoustic communication. We propose a decision feedback space receiving diversity equalizer embedded phase-locked loop in the uplink, and a transmitting diversity receiver for complex orthogonal space-time coding in the downlink.

We have made revisions to the introduction section and made clear descriptions of the contributions of this paper. Meanwhile, we have provided a comprehensive comparison with existing works in the literature on spatial diversity equalizers in underwater acoustics.

On page 2 line 42-57, add: “However, most communication technologies initially developed for terrestrial wired and wireless channels do not apply to the underwater acoustic environment [20]. In underwater acoustic communication, spatial diversity is also receiving significant attention as one of the techniques to improve performance under challenging channel conditions. For the joint reception of multiple signals in frequency-selective fading channels, antenna arrays and equalizers are used for diversity reception to improve the quality of wireless communication systems [21]. [22, 23] proposed multichannel DFE receiver for underwater acoustic communications. Received signals are processed by a bank of adaptive linear filters that jointly perform matched filtering and feed-forward equalization. Adaptive phase synchronization is then performed on each branch before the signals are combined and passed to a single DFE feedback and decision loop. In [24], the received signal is split into two streams. One stream is equalized after time reversal (TR) combining. Another stream is flipped in the time domain, processed by TR-combining and equalization, and then flipped in the time domain again. After processing, each stream is conducted diversity combining.”

On page 3 line 109-133, we have made revisions: “...The main contributions of this paper are as follows. 

1. The distributed... 2. To input different... 3. We propose a receiving... 4. We propose a transmitting...”.

On page 9 line 303-308, add: “Compared with the SIMO-DFE system described in [22, 23], this paper combines spatial diversity techniques with an adaptive decision feedback equalizer embedded PLL at the receiver. Phase-compensated and equalization are performed separately on each branch before diversity combining. Then the output of each equalizer is combined with the maximum signal-to-interference ratio criterion (MSIRC). The combined signal is decided and input into the FBF of each equalizer and the phase detector of each branch, rather than the combined signal itself.”

Reviewer#2, Concern # 5: Please add a list of Abbreviations (Acronym) after Conclusion Section.

Author response: Thank you for your valuable suggestion. We have added a list of Abbreviations after Conclusion Section on page 19.

Abbreviations 

DFE-PLL-MSIRC decision feedback equalization embedded phase-locked loop and maximum 

signal-to-interference ratio combining

DUA-DCS distributed underwater acoustic diversity communication system

COSTBC complex orthogonal space-time block code

DAS distributed antennas system

DFE decision feedback equalization

SNR signal-to-noise ratio

MRC maximal ratio combining

OFDM orthogonal frequency division multiplexing

TR time reversal

STBC space-time block code

OSTBC orthogonal space-time block code

GSTBC-SM generalized space-time block coded spatial modulation

MIMO multiple-input and multiple-output

SM spatial modulation

Q-OSTBC quasi-orthogonal space-time block code

STBC-NOMA space–time block coded non-orthogonal multiple access

UASNs underwater acoustic sensor networks

NSFC National Natural Science Foundation of China

PLL phase-locked loop

RF radio frequency

FFF feedforward filter

FBF feedback filter

RLS recursive least square

VCO voltage-controlled oscillator

SIMO single input multiple output

MSIRC maximum signal-to-interference ratio criterion

MSE mean square error

MISO multi-input single-output

BER bit error rate

Reviewer#2, Concern # 6: In the introduction, the authors did not give clearly describe the new methods and contributions that lead to improving the comprehensive survey of image enhancement methods compared to existing works. Please explain the new comprehensive contributions of this work (methodology or algorithm?) that will improve the image enhancement existing works more specifically.

Author response: Thank you for your valuable suggestion. In order to provide a clearer description of the novel algorithm and its contributions in this paper, we have made revisions to the introduction section. At the same time, I 'm sorry to make you confused. The main theme of this paper is not about image enhancement. We have added relevant references for spatial diversity equalizers for underwater acoustic communication. Additionally, we have provided a detailed description of the contributions of this paper.

On page 2 line 42-57, add: “However, most communication technologies initially developed for terrestrial wired and wireless channels do not apply to the underwater acoustic environment [20]. In underwater acoustic communication, spatial diversity is also receiving significant attention as one of the techniques to improve performance under challenging channel conditions. For the joint reception of multiple signals in frequency-selective fading channels, antenna arrays and equalizers are used for diversity reception to improve the quality of wireless communication systems [21]. [22, 23] proposed multichannel DFE receiver for underwater acoustic communications. Received signals are processed by a bank of adaptive linear filters that jointly perform matched filtering and feed-forward equalization. Adaptive phase synchronization is then performed on each branch before the signals are combined and passed to a single DFE feedback and decision loop. In [24], the received signal is split into two streams. One stream is equalized after time reversal (TR) combining. Another stream is flipped in the time domain, processed by TR-combining and equalization, and then flipped in the time domain again. After processing, each stream is conducted diversity combining.”

On page 3 line 109-133, the main contribution of the paper is revised as follows:

1. The distributed underwater acoustic diversity communication system (DUA-DCS) that we propose can solve the issue of unreliable point-to-point underwater acoustic communication links in shallow water environments and achieve a waveform-level diversity combination of underwater acoustic signals. 

2. To input different complex marine environment parameters into the bellhop model to obtain the underwater acoustic channel models under different shallow seabed topography and sediment environments and calculate the performance of the proposed system in these different marine environments. 

3. We propose a receiving diversity processing mechanism of joint decision feedback equalization embedded phase-locked loop and maximum signal-to-interference ratio combining (DFE-PLL-MSIRC), which exhibits the following advantages: 

• To eliminate the residual frequency and phase errors after Doppler compensation, a PLL module is embedded in each branch of the DFE.

• The combining coefficient of each branch adopts the maximum signal-to-interference ratio criterion, which considers the residual inter-symbol interference after equalization in addition to additive noise interference, leading to efficient and accurate computation. 

• The combined decision output is fed back to the feedback filters in each branch to ensure more accurate feedback output. 

•Considering system performance, complexity, and practical engineering costs, some engineering recommendations are provided.

4. We propose a transmitting diversity processing mechanism of complex orthogonal space-time block code (COSTBC). By utilizing a newly designed generalized complex orthogonal transmission matrix, complete transmission diversity can be achieved at the coding rate of 3/4.

Reviewer#2, Concern # 7: The authors mentioned only some related work, but without enough details. So it is necessary to add existing works in the related work section with the comparison table to improve this manuscript. Please revise it.

Author response: Thank you for your valuable suggestion. In order to provide a more detailed description of the related work mentioned in the manuscript, we have included the following content. Furthermore, we have added a comparison table of existing works in the appendix.

On page 1 line 2-8, add: “The breadth and depth of research on smart oceans and transparent oceans for ocean development and utilization have been continuously strengthened in recent years. By utilizing the features of broad coverage, fade resistance, and minimal required switching provided by the distributed antenna systems (DAS) [1, 2], underwater sensor nodes collect and transmit data through point-to-point and multipoint-to-multipoint communication to construct a distributed underwater sensor network for the effective transmission of information [3–6].”

On page 2 line 35-38, add: “In [18], the Maximum Ratio Combining (MRC) diversity algorithm is combined with the Orthogonal Frequency Division Multiplexing (OFDM) system to improve the received signal at the receiver. The primary idea behind MRC diversity is to improve the received signal at the receiver.”

On page 2 line 42-57, add: “However, most communication technologies initially developed for terrestrial wired and wireless channels do not apply to the underwater acoustic environment [20]. In underwater acoustic communication, spatial diversity is also receiving significant attention as one of the techniques to improve performance under challenging channel conditions. For the joint reception of multiple signals in frequency-selective fading channels, antenna arrays and equalizers are used for diversity reception to improve the quality of wireless communication systems [21]. [22, 23] proposed multichannel DFE receiver for underwater acoustic communications. Received signals are processed by a bank of adaptive linear filters that jointly perform matched filtering and feed-forward equalization. Adaptive phase synchronization is then performed on each branch before the signals are combined and passed to a single DFE feedback and decision loop. In [24], the received signal is split into two streams. One stream is equalized after time reversal (TR) combining. Another stream is flipped in the time domain, processed by TR-combining and equalization, and then flipped in the time domain again. After processing, each stream is conducted diversity combining.”

On page 20, add: S1 Appendix.

Reviewer#2, Concern # 8: There are too many spelling and grammar mistakes in the paper. It needs proper spelling and grammar checking.

Author response: Thank you for your valuable suggestion. We apologize for so many spelling and grammar mistakes. We have thoroughly reviewed the entire paper and made corrections to the spelling and grammar mistakes.

Reviewer#2, Concern # 9: Most mentioned Figures includes small details; so they are not clear for the readers to distinguish the difference and the performance improvement of image enhancement methods. Add full description for all the results.

Author response: Thank you for your valuable suggestion. We have added comparison experiments with the algorithm proposed in ref [22, 23] in both the simulation and lake trial sections. The simulation results are shown in Fig 13. And to provide a better description of the data in the figure, we have extracted the key data and presented them in the form of tables. At the same time, we have provided a detailed description of the results comparing our proposed algorithm with the results of the comparison algorithms.

On page 14 line 408-421:

“The simulation results are shown in Fig 13. The number of receiving antennas also represents the number of diversity.

From Fig 13, Table 3 and Table 4, it can be observed that the system’s BER decreases as the number of receiving antennas increases. Compared to the point-to-point communication link system, DFE-PLL-MSIRC can achieve a diversity gain of more than 5.2 dB, while the comparison algorithm can achieve a diversity gain of more than 2.3 dB. The proposed DFE-PLL-MSIRC can obtain about 3 dB more diversity gain than the comparison algorithm. When SNR is 0 dB, the BER of DFE-PLL-MSIRC can be reduced by at least one order of magnitude, while the comparison algorithm can be reduced by at least 0.4 order of magnitude. 

As seen from Fig 14, with the gradual deterioration of the marine environment, the BER of the same diversity also gradually increases. The minimum diversity order required for the system to maintain communication is shown in Table 5. We can observe that in the cases of marine environment 1 and 3, the comparison algorithm requires a higher diversity order than DFE-PLL-MSIRC, thereby validating that DFE-PLL-MSIRC is more efficient than the comparison algorithm.”

“It can be seen from Fig 18 and Fig 19 that the proposed system’s constellation diagrams are more convergent with the increase of the diversity number. The constellation map of DFE-PLL-MSIRC tends to approach the constellation points {1, 1}, {1, −1}, {−1, 1}, and {−1, −1} of the original QPSK signal, more so than the constellation map of the comparison algorithm. It can be seen from Table 7 that in the case of the transceiver spacing of 500m, compared with point-to-point communication, the BER of two-point diversity combination can be reduced by an order of magnitude, and the BER of three-point diversity combination can be reduced to 0. In the case of the transceiver spacing of 1000 m, compared with point-to-point communication, the BER of two-point diversity combination can be reduced by 0.5 orders of magnitude, and the BER of three-point diversity combination can be reduced by 1 order of magnitude. Regardless of whether the transmission distance is 500m or 1000m, the BER of DFE-PLL-MSIRC is lower than that of the comparison algorithm.”

Reviewer#2, Concern # 10: In conclusion, you can show briefly the percentage of improvement of image enhancement methods in terms of quantitative evaluation measures.

Author response: Thank you for your valuable suggestion. We have revised the conclusion section and provided a brief presentation of the improvement percentages of the proposed spatial diversity equalizer and the comparison algorithms.

On page 18 line 519-524, add: “Simulation and lake experiments results show that, compared to the point-to-point communication link system, DFE-PLL-MSIRC can achieve a diversity gain of more than 5.2 dB and obtain about 3 dB diversity gain more than the comparison algorithm. Moreover, the BER of DFE-PLL-MSIRC can be reduced by at least one order of magnitude, which is lower by at least 0.6 order of magnitude compared to the comparison algorithm.”

Reviewer#2, Concern # 11: What are the trends and future direction of applying those proposed method? Please give your suggestions.

Author response: Thank you for your valuable suggestion. We have added the discusses about the trends and future directions of applying the proposed method at the end of the conclusion.

On page 19 line 531-536, add: “Further research on the distributed processing mechanism of underwater acoustic signals for spatial information networks, effectively extending the coverage of spatial information networks to the maritime domain, and achieving global stereo coverage of the space-sky-ground-sea integrated space information network is our future trend and direction.”

Reviewer#2, Concern # 12: Add some recently papers (2020-2023) in the manuscript.

Author response: Thank you for your valuable suggestion. We updated the manuscript by adding new references between the years 2020 to date.

On page 1 line 3-8: add [3-6] in “By utilizing the features of broad coverage, fade resistance, and minimal required switching provided by the distributed antenna systems (DAS) [1] [2], underwater sensor nodes collect and transmit data through point-to-point and multipoint-to-multipoint communication to construct a distributed underwater sensor network for the effective transmission of information [3-6].”

On page 4 line 159-162: add [41] in “In most scenarios, the underwater acoustic channel can be considered a slowly varying multipath channel with coherence over time. The sound rays emitted from the source travel to the receiver through multiple routes, and the received sound field combines all the arriving rays [41].”

[3] Sathish K, Ravikumar C, Srinivasulu A, Rajesh A, Oyerinde OO, et al. Performance and Improvement Analysis of the Underwater WSN Using a Diverse Routing Protocol Approach. Journal of Computer Networks and Communications. 2022;2022.

[4] Sathish K, Hamdi M, Chinthaginjala R, Pau G, Ksibi A, Anbazhagan R, et al. Reliable Data Transmission in Underwater Wireless Sensor Networks Using a Cluster-Based Routing Protocol Endorsed by Member Nodes. Electronics. 2023;12(6):1287.

[5] Sathish K, CV R, Ab Wahab MN, Anbazhagan R, Pau G, Akbar MF. Underwater Wireless Sensor Networks Performance Comparison Utilizing Telnet and Superframe. Sensors. 2023;23(10):4844.

[6] Sathish K, Ravikumar CV, Rajesh A, Pau G. Underwater wireless sensor network performance analysis using diverse routing protocols. Journal of Sensor and Actuator Networks. 2022;11(4):64.

[41] Zhou M, Zhang H, Lv T, Li H, Xiang D, Huang S, et al. Underwater acoustic channel modeling under different shallow seabed topography and sediment environment. In: OCEANS 2022-Chennai. IEEE; 2022. p. 1–7.

---

## [Editor Report · Decision Letter 3]

7 Dec 2023

Spatial diversity processing mechanism based on the distributed underwater acoustic communication system

PONE-D-22-31101R3

DearAuthors,

We’re pleased to inform you that your manuscript has been judged scientifically suitable for publication and will be formally accepted for publication once it meets all outstanding technical requirements.

Kind regards,

Dr. Ravikumar Chinthaginjala, Ph.D

Academic Editor

PLOS ONE

Additional Editor Comments (optional):

Dear Authors,

Thank you for submitting Plosone Journal....All the best
---

## [Editor Report · Acceptance letter]

19 Dec 2023

PONE-D-22-31101R3 

PLOS ONE

Dear Dr. Zhou, 

I'm pleased to inform you that your manuscript has been deemed suitable for publication in PLOS ONE. Congratulations! Your manuscript is now being handed over to our production team.

Kind regards, 

on behalf of

Dr. Ravikumar Chinthaginjala 

Academic Editor

PLOS ONE